# Electronic structure of mononuclear and radical-bridged dinuclear cobalt(II) single-molecule magnets

David Hunger [1,10], Julia Netz[2,10], Simon Suhr[3,10],
Komalavalli Thirunavukkuarasu[4], Hans Engelkamp [5], Björn Fåk[6], Uta Albold[7],
Julia Beerhues[3], Wolfgang Frey[8], Ingo Hartenbach[3], Michael Schulze [9],
Wolfgang Wernsdorfer [9], Biprajit Sarkar [3,7] ✉, Andreas Köhn [2] ✉ &
Joris van Slageren [1] ✉

Metal-organic compounds that feature magnetic bistability have been proposed as bits for magnetic storage, but progress has been slow. Four-coordinate cobalt(II) complexes feature high inversion barriers of the magnetic moment, but they lack magnetic bistability. Developing radical-bridged polynuclear systems is a promising strategy to encounter this; however detailed investigations of such species are scarce. We report an air-stable radical-bridged dinuclear cobalt(II) complex, studied by a combination of magnetometry and spectroscopy. Fits of the data give $D = -113$ cm$^{-1}$ for the zero-field splitting (ZFS) and $J = 390$ cm$^{-1}$ for the metal–radical exchange. Ab initio investigations reveal first-order spin–orbit coupling of the quasi-degenerate d$_{x^2-y^2}$ and d$_{xy}$ orbitals to be at the heart of the large ZFS. The corresponding transitions are spectroscopically observed, as are transitions related to the exchange coupling. Finally, signatures of spin-phonon coupling are observed and theoretically analyzed. Furthermore, we demonstrate that the spectral features are not predominantly spin excitations, but largely vibrational in character.

Several decades ago, the idea arose that paramagnetic molecules could replace the magnetic particles used in hard disk storage devices, which could allow data densities that were orders of magnitude higher than the state of the art at the time. These investigations started with a dodecanuclear manganese cluster with an $S = 10$ ground state[1]. This ground state possesses a sizable zero-field splitting (ZFS), resulting in an energy barrier towards inversion of the magnetic moment, and concurrent slow relaxation at low temperatures[1]. Molecules that possess this property have been named single-molecule magnets (SMMs).

Because for the simplest case and an integer spin ground state, this energy barrier $U_{\text{eff}}$ is given by $U_{\text{eff}} = DS_z^2$ with the axial ZFS constant $D$ and the ground state spin $S$, much energy was devoted to increasing $S$. As a result, a great deal of knowledge and understanding of the magnetic properties of high-spin clusters with large magnetic anisotropies, and especially of the interplay between the local ZFS and the isotropic exchange interactions $J$ between the paramagnetic ions were gained. Most data were interpreted in the strong-exchange approximation, i. e., assuming $J \gg D$, and the limitations of this model were explored[2].

[1]Institute of Physical Chemistry, University of Stuttgart, Stuttgart, Germany. [2]Institute of Theoretical Chemistry, University of Stuttgart, Stuttgart, Germany. [3]Institute of Inorganic Chemistry, University of Stuttgart, Stuttgart, Germany. [4]Department of Physics, Florida A&M University, Tallahassee, FL, USA. [5]HFML-FELIX, Nijmegen, The Netherlands. [6]Insitut Laue-Langevin, Grenoble, France. [7]Institute of Chemistry and Biochemistry, Freie Universität Berlin, Berlin, Germany. [8]Institute of Organic Chemistry, University of Stuttgart, Stuttgart, Germany. [9]Physikalisches Institut, Karlsruhe Institute of Technology, Karlsruhe, Germany. [10]These authors contributed equally: David Hunger, Julia Netz, Simon Suhr. ✉e-mail: biprajit.sarkar@iac.uni-stuttgart.de; koehn@theochem.uni-stuttgart.de; slageren@ipc.uni-stuttgart.de

Because $D$ decreases with increasing $S^3$, this approach was doomed to fail in terms of the envisaged practical application. Hence, attention turned to increasing the magnetic anisotropy. One solution was found in employing lanthanide ions, especially dysprosium(III). Here, the magnetic anisotropy originates from the crystal field splitting of the ground multiplet[4], resulting in energy barriers up to ca. 2000 K[5–8]. An alternative approach is the usage of low-coordinate transition metal ions[9], where spectroscopically certified energy barriers of up to 650 K have been reported[10]. In this respect, four-coordinate cobalt(II) complexes deserve special attention. On the one hand, these complexes are usually chemically quite robust, and on the other hand, large effective energy barriers and slow relaxation of the magnetization at high temperatures have been reported[11,12]. Four-coordinate cobalt(II) ions possess orbitally non-degenerate spin quartet ground states that are split into two Kramers doublets (KDs) by the ZFS. In spite of the promise of certain mononuclear lanthanide and transition metal complexes as SMMs, true magnetic bistability has been rare. Careful crystal field engineering by tailoring the ligand sphere has allowed sizable coercivities in lanthanides[5], but this approach is not necessarily robust, when considering that practical application will entail surface immobilization, with ensuing potential structural distortion. Developing exchange-coupled dinuclear complexes of highly anisotropic ions has proven to be a second viable strategy to engender magnetic bistability in lanthanide complexes[13], but this strategy is less common in transition metal chemistry[14,15]. One challenge here is that the exchange interactions should not be weaker than the local anisotropy, lest excited spin states are generated, which can serve as intermediate states in Orbach relaxation processes, lowering the effective energy barrier. In previous work, we have shown that going from the mononuclear [Co(bmsab)]$^{2-}$ to the radical-bridged dinuclear complex [{Co(H$_2$tmsab)}$_2$($\mu$-tmsab)]$^{3-}$ slows down the magnetization relaxation rate by a factor of up to 350[15]. Here bmsab is the dianion of 1,2-bis(methanesulfonamide) benzene and tmsab is the radical trianion of 1,2,4,5-tetrakis(methanesulfonamide)benzene. Fits of the magnetic data suggested the presence of both strong local ZFS, as well as strong exchange interactions. However, the complex could only be isolated through "crystal picking", resulting in extremely low yields. Therefore,

the energy spectrum was not studied in detail, precluding any detailed insight into the nature and origin of the magnetic interactions. We thus set out to develop a more convenient synthesis method, yielding a different, but related air- and moisture-stable radical-bridged dinuclear cobalt complex that can be prepared in satisfactory yields and quantities.

The energy splittings that are of relevance to cobalt-based single-molecule magnets lie in the THz-to-Far-Infrared regime. Consequently, a range of experimental techniques has been used to experimentally determine these splittings, especially those related to ZFS[16]. To study the lowest Kramers doublet and the ZFS for weakly anisotropic systems, high-frequency electron paramagnetic resonance (HFEPR) spectroscopy has been used[17–22]. A much-used and powerful method is variable-field far-infrared (FIR) spectroscopy, which allows direct determination of the energy gap between the two Kramers doublets[19,21–29]. Furthermore, inelastic neutron scattering (INS) also allows access to this energy gap, and the $Q$-dependence of the signal provides a means other than applying a field to distinguish between phonon and magnetic transitions[19,24]. Finally, also variable-field Raman spectroscopy has turned out to be a very powerful method in this regard[23,27]. In some cases, clear signatures of spin–phonon coupling were observed, i.e., excitations having combined spin and vibrational character[21,23–25,27]. Spin–phonon coupling is at the heart of the relaxation of the magnetic moment in single-molecule magnets[30,31]. Similar studies have been carried out on lanthanide-based single-molecule magnets[32–35].

Here we present a detailed experimental and theoretical study of the low-energy electronic structure of a previously reported mononuclear cobalt(II) single-ion magnet [K(18-crown-6)]$_2$[Co(bmsab)$_2$] (**1**)[36], and a radical-bridged dinuclear cobalt(II) complex, namely [K(18-crown-6)$_3$[Co(bmsab)]$_2$($\mu$-tmsab)] (**2**), see Fig. 1. The energies of low-lying excited states are experimentally determined by a combination of variable-field FIR and Raman, as well as inelastic neutron scattering spectroscopies. Through in-depth theoretical analysis, we demonstrate that the large single-ion ZFS is due to the strong mixing of the ground and low-lying excited quartet states of the individual ions by spin-orbit coupling. We determine the energies of the relevant excited

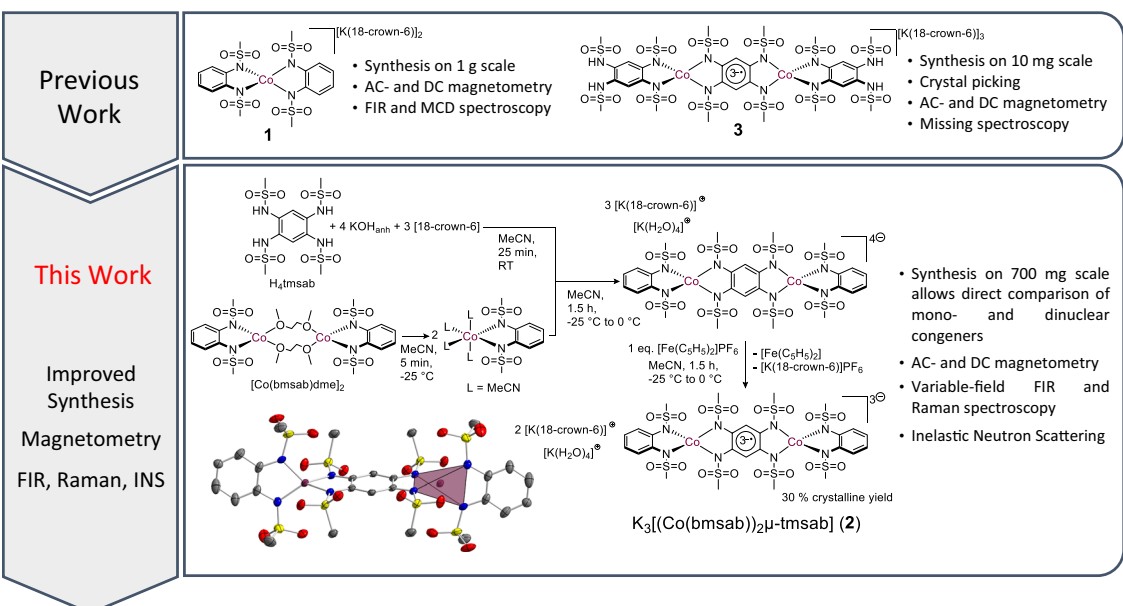

**Fig. 1 | Synthetic outline of the investigated molecules.** Top: In previous work, complexes **1**[32,36] and **3**[15] were studied separately by different techniques. Bottom: Schematic overview of the synthetic route to obtain **2**. This work provides a comprehensive spectroscopic and theoretical study on both **1** and the dinuclear complex **2**. In the bottom panel, the molecular structure of **2** as obtained from X-ray diffractometry is shown. The polyhedron highlights the distorted tetrahedral coordination geometry around the metal center. Ellipsoids are drawn at 50 % probability. H atoms and counter ions are omitted for clarity.

quartet states spectroscopically. In the dinuclear complex **2**, we find transitions with excitation energies attributable to local ZFSs, to exchange interactions and to local excited states. For both complexes, taking into account the local quartet excited states of the cobalt ions is essential for a theoretical description of the spectra. We find clear signatures of spin-phonon coupling and model the effects of this coupling on the spectra. This work reveals the reason behind the success of four-coordinate cobalt(II) as building blocks in single-molecule magnets and elucidates important theoretical and experimental aspects of spin-phonon coupling. Both will allow development of single-molecule magnets with improved properties.

## Results and Discussion
### Synthesis and structure
The in-depth investigation of magnetic materials, particularly with respect to spin-phonon coupling, requires significant amounts of phase-pure material. The previously reported synthesis of mononuclear cobalt(II) complex **1** allows its isolation on a gram-scale[36], enabling physical investigations that require such large amounts of sample, such as INS. While the structural motif of a radical bridged dinuclear Co(II) compound based on **1** was reported before in the complex [K(18-crown-6)]₃[Co(H₂tmsab)]₂($\mu$-tmsab)] (**3**, see Fig. 1, top)[15], its synthesis gave yields on the milligram-scale only. Hence, opportunities to conduct thorough physical studies were limited, precluding fundamental understanding of the electronic structure and possible spin-phonon interactions. We now present a synthetic procedure to obtain sufficient amounts of phase-pure, radical bridged dinuclear Co(II) compound **2**, whose structure is even closer to the mononuclear congener **1** (see Fig. 1, bottom). Two salient features of this procedure are (i) the use of well-defined, heteroleptic synthons and ii) the exploitation of cation-$\pi$ interactions to facilitate homogeneous crystal growth.

We recently reported heteroleptic complexes that can function as precursor molecules for polynuclear compounds[37] and showed their ability to serve as building blocks for radical-bridged, dinuclear complexes with four-coordinate metal centers[38]. Treatment of such a precursor molecule, [Co(bmsab)(dme)]₂ (dme = dimethylether), with deprotonated 1,2,4,5-tetrakis(methanesulfonamido)-benzene, followed by oxidation with ferrocenium, allowed the isolation of dinuclear **2** (see Fig. 1). After crystallization, phase-pure material was obtained in yields of up to 30%, allowing the isolation of several hundreds of milligrams of **2**. The chemical purity of the material was established by means of elemental analysis, UV/Vis/NIR spectroscopy (Fig. S1) and powder X-ray diffractometry (Fig. S2). The use of anhydrous KOH, and of 18-crown-6 as an encapsulating agent during deprotonation were instrumental in isolating crystalline material in a reproducible manner.

Compound **2** crystallizes in the triclinic space group $P\bar{1}$ (Table S1). The complex anion lies on a special Wyckoff position, with the inversion center located in the middle of the bridge, rendering the complex anion inversion-symmetric. The two capping ligands are oriented in a perpendicular fashion to the bridging ligand. The coordination geometry around each Co(II) center is of particular interest since this was found to be decisive for the magnitude and the sign of the axial ZFS parameter $D$[25,36]. For **2**, just as in the mononuclear compound **1**, a distorted tetrahedral coordination geometry with N−Co−N angles between 79.8(2)° and 133.3(2)° is found, giving distortion values of $\tau_4 = 0.73$ and $\tau_\delta = 0.68$[39,40], which are similar to the those found for previously reported **3**[15,36]. Using the coordination polyhedron of **1** as a reference for a SHAPE geometry analysis[41], a very low distortion value of 0.037 is found for **2** (compare Fig. 1, bottom). Consequently, an equally large, negative $D$ as for **1** can be expected in **2**. Inspection of the bond lengths provides insight into the charge states of the ligands. The bond lengths in the capping ligands of **2** are similar to the values observed in the dianionic bmsab²⁻ ligands in **1**. In **2**, the Co−N distances are shorter for the capping ligands (1.975(3) and 1.984(3) Å) than for

the bridging ligands (1.999(3) and 2.005(3) Å). This indicates a reduced donor strength of the bridge. Additionally, the C−N bond lengths in the bridging ligand are contracted (1.371(5) and 1.378(5) Å) compared to the capping ligands (1.406(5) and 1.414(6) Å). See Supplementary Information Figs. S3–S6 and Table S2 for additional information.

These geometrical parameters indicate a dianionic state of the capping ligands and a trianionic, radical state of the bridge. The charge of the ligands is compensated by the two Co(II) ions and six K⁺ ions, which are shared between two neighboring molecules. The K⁺ ions are part of two different structural motifs in the extended structure of **2** (see Supplementary Information Fig. S4). In one dimension, the K⁺ ions are axially coordinated by the oxygen atoms of the sulfonyl groups, and equatorially by 18-crown-6. In the second dimension, the K⁺ ions form a sandwich-like structure with the aryl backbones of the capping ligands. Finally, four water molecules complete the coordination sphere of the K⁺ ions. These two different connection motifs lead to an interwoven 2D sheet of perpendicular strands in the extended crystal structure of **2**, which is a remarkable difference to **3**, where only single strands were observed[15]. Consequently, this suggests the importance of the $\eta^2$-coordination of the K⁺ ions as a design criterion for stable ionic networks that allows improved crystal growth in comparison to unconnected structures. This observation is in line with our recent results on the analogous dinuclear Ni(II) and Zn(II) complexes[38].

### Magnetic susceptibility
Magnetic measurements give first insight into the relevant magnetic parameters of the compounds. For mononuclear **1**, the room temperature susceptibility-temperature product is $\chi T = 3.41$ cm³ K mol⁻¹ [36]. Upon lowering the temperature, $\chi T$ remains essentially constant down to 100 K and then decreases slowly, before decreasing more strongly below 3 K. For the dinuclear complex **2**, the room temperature susceptibility-temperature product is $\chi T = 6.99$ cm³ K mol⁻¹, which is the value expected for a $S = \frac{5}{2}$ system with $g_{iso} = 2.53$ on the basis of the Curie-law (Fig. 2a). Upon lowering the temperature, a broad maximum is found in $\chi T$ at around 100 K ($\chi T = 8.6$ cm³ K mol⁻¹), and below 8 K, $\chi T$ drops drastically. To facilitate the interpretation of the magnetic properties of **2**, we also plot in Fig. 2a two times the $\chi T$ value for **1**, to which a constant value of 0.375 cm³ K mol⁻¹ is added to account for the contribution of the unpaired electron of the bridging ligand to the susceptibility (Fig. 2a). The $\chi T$ value for **2** is the same or slightly lower at room temperature compared to this reference value, but higher at all other temperatures. This observation is consistent with antiferromagnetic interactions between the $S = \frac{3}{2}$ cobalt spins and the bridging $S = \frac{1}{2}$ radical spin, leading to a high-spin ferrimagnetic ground state with $S = \frac{5}{2}$. The precipitous drop in $\chi T$ towards the lowest temperature is likely an effect of the relaxation time of the magnetic moment becoming slow on the measurement time scale.

To obtain a first estimate of the spin Hamiltonian parameters for **2**, we have fitted the $\chi T$ product of **2** to the following spin Hamiltonian:

$$\hat{H}_{d,1} = J(\hat{\mathbf{S}}_1 \cdot \hat{\mathbf{S}}_b + \hat{\mathbf{S}}_2 \cdot \hat{\mathbf{S}}_b) + D(\hat{S}_{z,1}^2 + \hat{S}_{z,2}^2) + \mu_B \mathbf{B} \cdot \mathbf{g} \cdot (\hat{\mathbf{S}}_1 + \hat{\mathbf{S}}_2) + \mu_B \mathbf{B} \cdot \mathbf{g}_b \cdot \hat{\mathbf{S}}_b, \tag{1}$$

where **B** is the magnetic field, $\hat{\mathbf{S}}_i = (\hat{S}_{x,i}, \hat{S}_{y,1}, \hat{S}_{z,i})$ the spin operator of the respective spin center (1 and 2 for the two cobalt ions, $b$ for the radical bridge), and $\mu_B$ the Bohr magneton. We do not consider second-rank transverse ZFS, which previous work revealed to be negligibly small[25,36]. This assumption is corroborated by the theoretical calculations presented below. In line with the inversion symmetry of the compound, the same Zeeman tensors **g** and axial ZFS parameters $D$ are used for both metal centers, as well as the same isotropic exchange coupling parameter $J$ for the interaction of each Co(II) ion with the radical bridge. The $g$-tensor of the radical bridge is denoted $\mathbf{g}_b$ and will be assumed to be isotropic with a fixed value of 2.0. The best

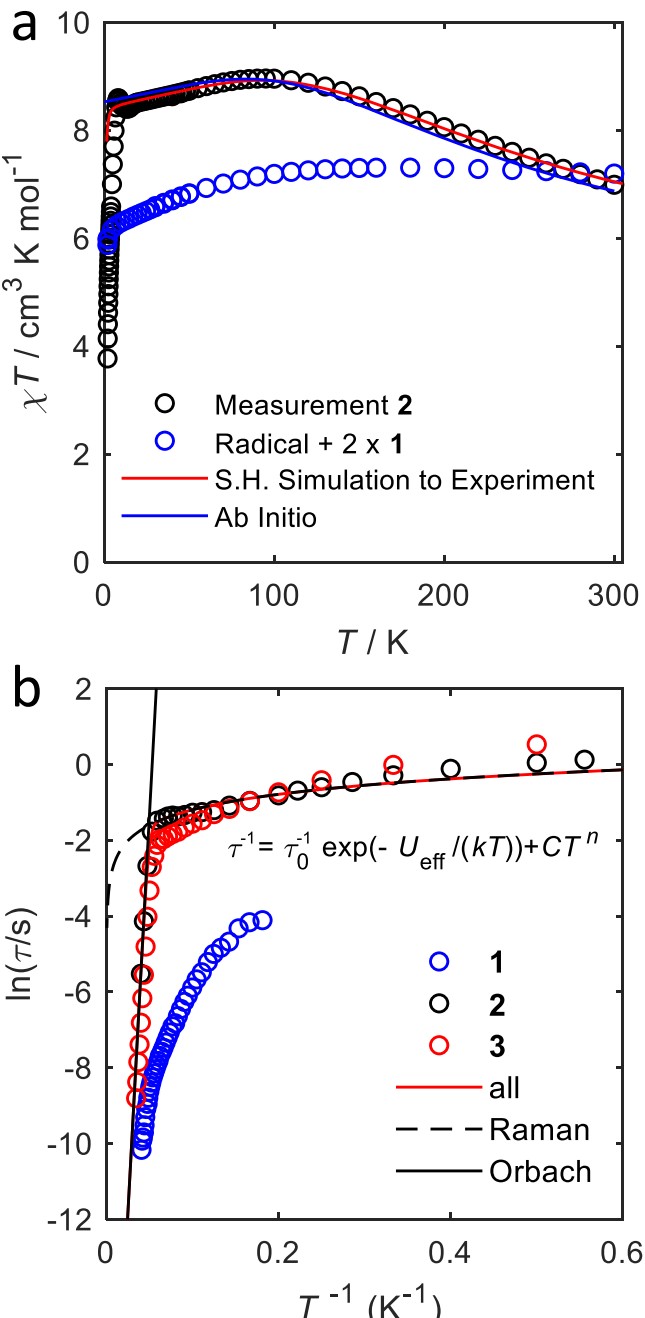

**Fig. 2 | Static and dynamic magnetometry measurements. a** Measured temperature dependence of the the temperature-susceptibility product of **2** (black circles) in comparison with the sum of two times the values for the mononuclear species **1** plus a contribution $\chi T = 0.375\ \text{cm}^3\ \text{K mol}^{-1}$ for the radical bridge (blue circles). The corresponding spin Hamiltonian simulations and ab initio calculated curves are shown as red and blue lines, respectively. **b** Natural logarithm of the relaxation time $\tau$ of **1**[25], **2** and **3**[15] in dependency of the inverse temperature at zero external field. Data is shown as open circles and a fit to the experimental data of **2** based on the equation given in the figure as a solid red line. The single components are shown as black solid or dashed lines.

agreement between experiment and simulation is obtained by using an axial ZFS value of $D = -145(29)\ \text{cm}^{-1}$, as well as principal values of the g-tensor of $g_x = 2.13(14)$, $g_y = 2.13(14)$ and $g_z = 3.1(1)$ for the two Co(II) centers, and an antiferromagnetic exchange coupling parameter of $J = 322(88)\ \text{cm}^{-1}$. Magnetization measurements at different temperatures can be simulated in reasonable agreement with the experiment with

the same parameter set (Fig. S7). Nevertheless, simulation of the magnetization deviates from the low temperature magnetization measurement results (1.8 and 5 K), due to slow magnetization dynamics (see below).

Compounds **1** and **2** also display ac magnetic susceptibility behavior consistent with single-molecule magnetism (Section S3). The dynamic properties of **1** have been reported previously[25,36]. **2** shows a relaxation time of 1.1 s at 1.8 K in zero applied magnetic field, which is well in line with the previously reported relaxation times of **3**[15]. In zero field, a clear, non-zero out-of-phase susceptibility signal $\chi''_{mol}$ is measured for **2** up to 25 K. The temperature dependence of the zero-field relaxation times $\tau$ for **2** was fitted to a sum of an Orbach and a (pseudo) Raman process (Fig. 2b). On this basis, values of $\tau_0 = 2.13(3) \cdot 10^{-10}$s, $U_{eff} = 291(1)\text{cm}^{-1}$, $C = 0.85(8)\text{s}^{-1}\text{K}^{-n}$ and $n = 0.59(4)$ were obtained. The parameter values found are similar to the reported ones for **3**[15], which shows that dinuclear **2** features equally improved relaxation characteristics compared to mononuclear **1**. Since an exponent of $n = 0.59(4)$ is much too low for a Raman process[25], the observed low-temperature relaxation cannot be of purely Raman-like nature, but can be best described as a pseudo Raman relaxation which shares similarities to a quantum tunneling mechanism (QTM). In order to investigate this, ac susceptibility measurements in applied external magnetic fields of 500 Oe up to 10,000 Oe were carried out (compare Section S3), as well as dc relaxation measurements were performed (Figs. S11 and S12). Such an application of a magnetic field can suppress QTM and hence an increase in $\tau$ is expected if the main relaxation pathway at low temperatures is related to QTM. Indeed, as soon as a small magnetic field of 500 Oe is applied, a clear increase in $\tau$ is observed, with a further increase in higher dc fields. Fits of the relaxation times as a function of the temperature at different applied fields on the basis of the equation shown in Fig. 2b reveal an increase of the Raman exponent with increasing applied magnetic fields from 0.59 at 0 Oe via 0.95 at 500 Oe and 2.4 at 2000 Oe to around 3 above 4000 Oe, before a slight decrease to 1.4 at 10,000 Oe (see Fig. S13, Table S3). These observations reveal that also **2** suffers from QTM at zero applied magnetic field. In order to investigate the relaxation dynamics of **2** further, hysteresis measurements were carried out at temperatures from 1.8 K up to 15 K (Fig. S14), and at temperatures from 30 mK up to 5 K by micro-SQUID (Figs. S15 and S16)[42]. While hysteresis curves at 1.8 K of **1** have already been reported[25,36], an investigation of the magnetic hysteresis at temperatures below that has not been carried out. Hence, the mK measurements on **2** are accompanied by a measurement series on **1**, where the mononuclear complex can again serve as an adequate comparison for the dinuclear one **2** (Figs. S17 and S18). For both compounds, at all temperatures a waist-restricted hysteresis is observed, which demonstrates efficient relaxation around zero field, well in line with the behavior observed in ac susceptibility. In the case of **2**, a small coercivity can be seen (Fig. S15).

Although the least-squares fit of $\chi T$ to the spin Hamiltonian of Eq. (1) gives first insight into the electronic structure of **2**, it also reveals a large covariance of the parameters $D$ and $J$ and a sizeable error bar for both parameter values. Hence, the magnetometry experiments alone do not give sufficient physical insight into the system, which prompted us to tackle this system by a combined spectroscopic and theoretical approach.

## Far-Infrared and Raman measurements

Deeper experimental insight into the spin structure of the two compounds **1** and **2** can be obtained from spectroscopy. To this end, electron paramagnetic resonance (EPR) spectroscopy can be expected to be an excellent tool. Unfortunately, the two Co(II) compounds presented in this work are EPR silent, because the transition within the lowest Kramers doublet (KD) is formally forbidden, while the allowed inter-KD transitions are in the far-infrared (FIR) regime and hence at much higher energies ($|2D| \approx 250\text{cm}^{-1}$) than conventional microwave

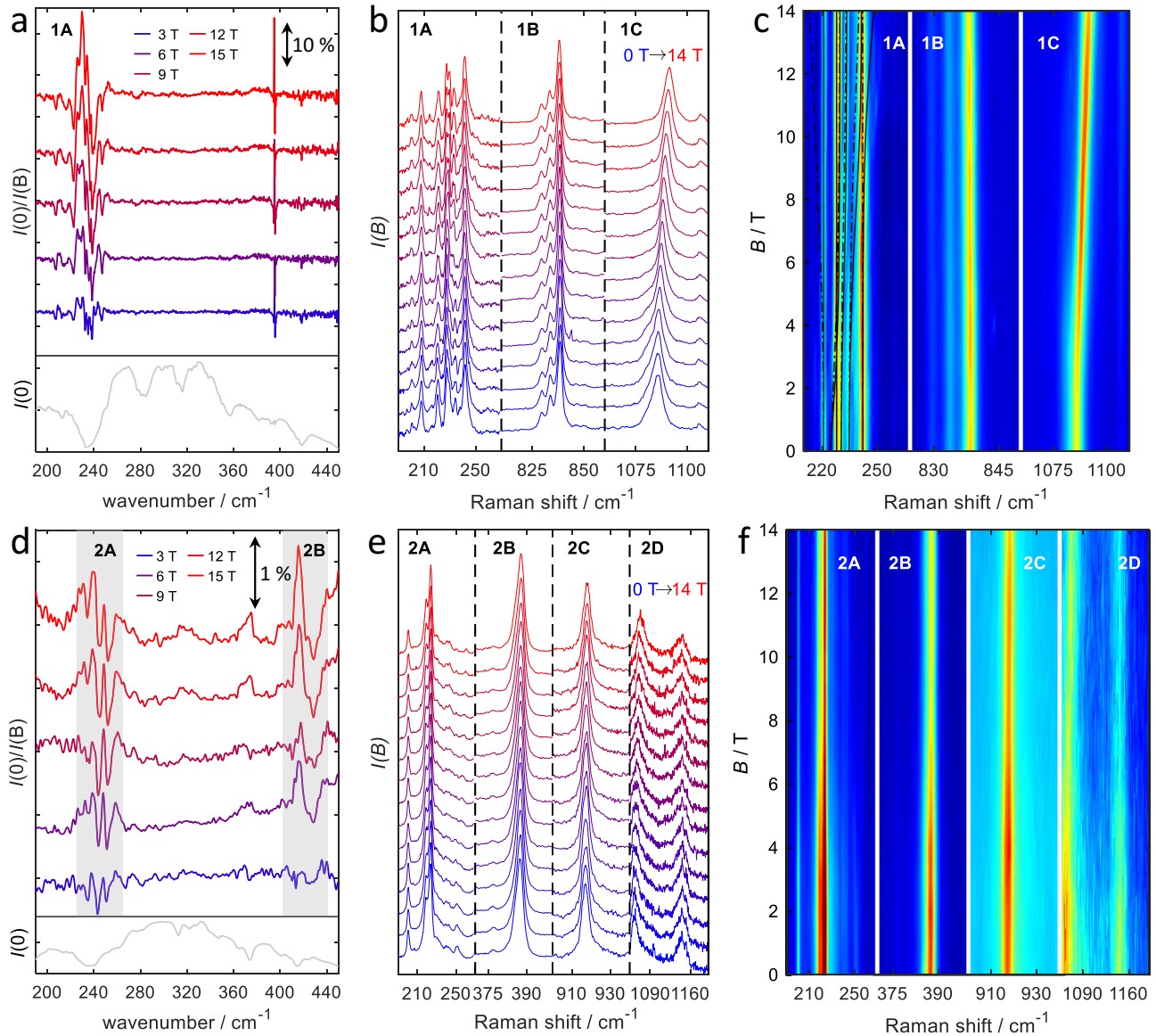

**Fig. 3 | Overview of the variable-field spectra recorded on 1 and 2.** FIR spectra of **1** (**a**) and **2** (**d**), recorded at 5 K and different fields as indicated in the figure. Spectra were normalized by dividing the zero-field signal intensity by the in-field signal intensity. As a consequence, zero-field features are pointing down. Note the different y-scales for both compounds. The feature observed at *ca*. 390 cm⁻¹ for **1** is a measurement artifact. b,e: Raman spectra of **1** (**b**) and **2** (**e**), recorded at 2 K, and different fields as indicated in the figure. False-color plots of the variable-field Raman spectra recorded on **1** (**c**) and **2** (**f**). Signal **1A** is overlaid with the energies of the calculated spin-phonon coupled transitions.

sources cover. Therefore, variable-field far-infrared (FIR) and Raman spectra were recorded for both **1** and **2** (Fig. 3, Table 1). Both techniques are complementary to each other since Raman and FIR follow different selection rules, and also because, as it will turn out (see below), the Raman spectra can be recorded up to higher excitation energies.

**Mononuclear compound 1.** FIR spectra were recorded on a pressed pellet of **1** at 3 and 5 K in the range between 30 cm⁻¹ and 680 cm⁻¹ and at fields from 0 to 30 T (Fig. 3, Table 1; for a complete field overview see Supplementary Information Fig. S19). In order to bring field-dependent features to the fore, the in-field FIR spectra were normalized with respect to the zero-field spectrum by division of the transmission signal of the zero field spectrum by that of the in-field spectra (for a detailed description see Supplementary Information section S4). This means that positive features indicate *increased* absorption in the field compared to the zero field. In these spectra, a series of peaks

(denoted **1A**) is observed in the region around 230−240 cm⁻¹, where, in view of the fits of the magnetometry data, only the excitation from the ground to excited KD, with energy 2$D$ is expected. The presence of several peaks in this region had been noted before for this[36], and related complexes[25], and this finding was attributed to spin−phonon coupling, but not investigated in detail. In the variable-field Raman spectra (Figs. S20, S21, S22), several apparently field-dependent features are observed in this area as well, in particular, strongly field-dependent features at 230 and 234 cm⁻¹ and a less field-dependent peak at 242 cm⁻¹. It should be noted that the Raman spectra were obtained by excitation at 532 nm. At this wavelength, the Co complex shows a strong absorption and we can therefore assume that near-resonant Raman spectra are recorded, which particularly enhances the signals of local vibrations of the anion. The shift of the peak frequency with field is rather small, compared to what would be expected for the Zeeman splitting of a pure magnetic resonance transition. Interestingly, further peaks are visible in the Raman spectrum at around 835

**Table 1 | Transition energies observed in the FIR, Raman and INS spectra of 1 and 2, together with *abinitio* calculated values**

| Compound | Transition | FIR | | Raman | | INS | | ab initio |
|---|---|---|---|---|---|---|---|---|
| | | $E/cm^{-1}$ | Type | $E/cm^{-1}$(0 T) | Type | $E/cm^{-1}$ | Type | $E/cm^{-1}$ |
| 1 | 1A | 216–252 | m | 217–248 | m | 240–265 | m | 253 |
| | 1B | – | – | 826 - 843 | m | – | – | 847 |
| | 1C | – | – | 1085 | s | – | – | 1160 |
| 2 | 2A | 222–267 | m | 200 - 257 | m | 242 | s | 268 |
| | 2B | 416 | s | 387 | s | 363 | s | 386 |
| | 2C | – | – | 917 | s | – | – | 1191 |
| | 2D | – | – | 1063 | s | – | – | 1339 |

Type: s single peak, m multiple peaks.

cm$^{-1}$ (**1B**), and around 1075 cm$^{-1}$ (**1C**). The energy of the former feature is rather independent of magnetic field strength. In contrast, the latter signal has a stronger field dependence consistent with a pure magnetic resonance transition.

**Dinuclear compound 2**. Analogously to **1**, FIR and Raman spectra were measured in different applied fields. The FIR spectra of **2** reveal two distinct field-dependent features (Fig. 3d). The first of these is found at 240 cm$^{-1}$ (**2A**), and the second is observed in the region of 420 cm$^{-1}$ (**2B**). Analogously to what was found for **1**, it can be expected that the transition frequency of peak **2A** corresponds to the expected energy gap between ground and excited KDs. A similar structure to what was observed for **1** is observed here for **2**. At the position of the second field-dependent peak (**2B**) found for **2**, nothing can be observed in the FIR spectra recorded on **1**, and it is tempting to link the corresponding excitation to the exchange coupling between metal and radical spins. In the variable-field Raman spectra for **2** (Figs. 3, S23, S24), features are found at similar but slightly lower frequencies, indicating a more complex origin of these peaks than simple magnetic resonance or electronic transitions. In addition, the Raman spectra show two further peaks at higher energies of 917 cm$^{-1}$ (**2C**) and 1063 cm$^{-1}$ (**2D**), again reminiscent of what was found for compound **1**.

**Inelastic neutron scattering**
High-energy transitions in SMMs can also be probed by inelastic neutron scattering (INS). While for FIR and Raman spectroscopies, a magnetic field is applied for an experimental distinction between phonon modes and spin transitions, this is typically not done in the case of INS. Instead the dependence on the transferred momentum $Q$, as well as the temperature dependence of the intensities of the observed features can serve the same purpose. It is of note that the highest energies where transitions have been observed in INS on SMMs are currently at around 6 meV (50 cm$^{-1}$)[24]. Hence, the present work provides an opportunity to exploit the limits of INS in the context of SMMs and it is of great interest whether and how the transitions **1A, 2A** and **2B** can be investigated by means of neutron scattering.

INS measurements were carried out on powdered samples of **1** and **2** at the PANTHER instrument at the Institute Laue-Langevin. Compound **1** was measured at temperatures of 1.5, 50 and 100 K with $E_i$ = 50 meV; in the case of **2**, a more extensive temperature study was carried out with eight temperatures in the range from 1.5 K to 200 K at $E_i$ = 76 meV. The scattering function $S(Q, E)$ for **1** and **2** recorded at $T$ = 1.5 K is shown in Fig. 4; spectra recorded at other temperatures can be found in the Supplementary Information (Figs. S26, S27).

In the INS spectra recorded on **1**, four features are observed. The energetically lowest three are located at 4.8 meV (39 cm$^{-1}$), 12.5 meV (100 cm$^{-1}$), and 18.9 meV (152 cm$^{-1}$), respectively, and overlap partially. A fourth feature has components at 30 meV (240 cm$^{-1}$) and 33 meV (265 cm$^{-1}$). The energy of this fourth feature is close to that of transition **1A**. The INS spectrum recorded on **2** at 1.5 K is similar to that of **1**. In

the low-energy regime, overlapping peaks can be observed at around 10 meV (80.5 cm$^{-1}$) and at 20 meV (161 cm$^{-1}$). Furthermore, two isolated peaks are found, with the more intense one at 30 meV (242 cm$^{-1}$) and the weaker one at 45 meV (363 cm$^{-1}$). While the peak at 30 meV (242 cm$^{-1}$) lies at similar energies to transition **2A**, the one at 45 meV (363 cm$^{-1}$) lies at slightly lower energies to **2B**. Overall, the transition energies agree well with those observed in FIR and Raman (Table 1).

In Figs. S28 and S29, we plot the spectra as a function of temperature. For vibrational transitions, no major temperature dependence is expected, while magnetic transitions will decrease in intensity. For **1** the first three signal branches show an increase in $S(Q, E)$ from 1.5 K to 100 K, especially due to the increase of quasi-elastic background, indicating their vibrational origin. Interestingly, the fourth peak at around 30 meV (240 cm$^{-1}$) decreases in intensity with increasing temperatures. This behavior is well in line with a magnetic transition corroborating its assignment to transition **1A**. Similarly, for **2** the two high-energy features at 30 meV (242 cm$^{-1}$) and 45 meV (363 cm$^{-1}$) decrease in intensity towards higher temperatures suggesting their magnetic origin. Interestingly, the 30 meV (242 cm$^{-1}$) signal decreases monotonically with increasing temperature, while the the intensity of the peak at 45 meV (363 cm$^{-1}$) increases again from 100 K upwards.

In order to separate the magnetic transitions from the phonon background, the spectra were corrected by a Voigtian baseline and fitted by a sum of Gaussians (for details see Figs. S28 and S29). In this context, the temperature dependence of the area of each Gaussian peak is of great interest (compare Fig. 4b, e). Similarly to the FIR and Raman measurements of **1**, the feature due to the **1A** transition also displays some structuring. For **2**, the transition **2A** is not structured, but shows a small shift towards lower energies at higher temperatures. Furthermore, the fitted intensity for the peak at 45 meV only displays a monotonic decrease of $S(Q, E)$ from 1.5 K to 200 K. Consequently, the increase of the signal from 100 K upwards in the raw experimental data is due to the increase in phonon background. Nevertheless, it is of note that the Gaussian curves that are attributed to the transitions **1A, 2A** and **2B** show a distinct decrease of their area with increasing temperatures. This behavior strongly suggests a magnetic origin of these experimental features. In summary, the peaks measured in INS are in good agreement with the transitions that were observed in FIR and Raman, but the larger line widths in INS obscure some of the fine structure found in the other spectra.

**Theoretical modeling of electronic structure**
To gain more insight into the measured spectra, detailed computations of the electronic structure of the anions **1** and **2** proved to be indispensable. We carried out ab initio computations using complete active space self-consistent field (CASSCF) and CAS second-order perturbation theory (CASPT2) methods, followed by spin-orbit configuration interaction (SO-CI) computations. These computations were based on the crystal structures, reoptimized at the density functional theory (DFT) level with periodic boundary conditions. We also

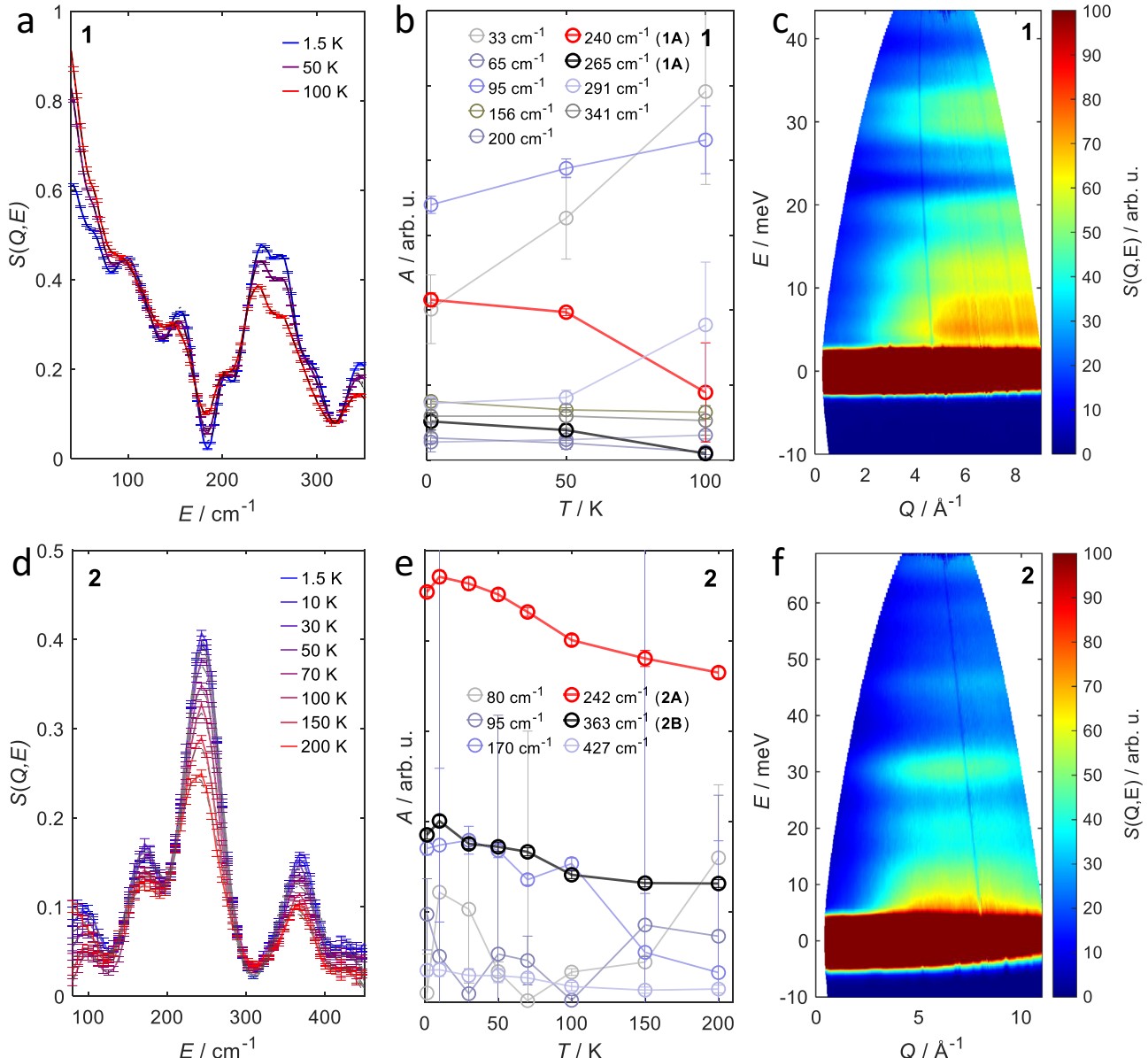

**Fig. 4 | Overview of the inelastic neutron scattering experiments.** Baseline corrected experimental $S(Q,E)$ of **1** (**a**) and **2** (**d**) at a fixed $Q$ of $Q = 3.0 \pm 0.5\,\text{Å}^{-1}$ at the indicated temperatures (colored, solid lines). The corresponding fits based on a sum of Gaussians are pictured as grey dashed lines in the left plots. Deconvolutions of the Gaussians into the respective components are found in Figs. S28 and S29.

**b, e** Area of the single Gaussian peaks against the temperature. The energy of each Gaussian is highlighted in the legend. **c, f** Scattering function $S(Q,E)$ of **1** (**c**) and **2** (**f**) as a function of energy ($E$) and momentum transfer ($Q$) at a temperature of 1.5 K. Measurements were carried out on non-deuterated, powdered samples at incident energies of $E_i = 50\,\text{meV}$ for **1** and $E_i = 76\,\text{meV}$ for **2**.

performed DFT optimizations of the isolated complex ions, which did not lead to any significant change of the structures. Further details of the computations are given in the Experimental Section and in the Supplementary Information Section S6. The optimized geometries as well as the computational results are summarized in the Supplementary Data 1 and the Supplementary Data 2.

The resulting lowest energy levels up to 1400 cm⁻¹ are shown in Fig. 5 together with the corresponding magnetic moment expectation values $\langle \mu_z \rangle$ along the main magnetic axis. The latter coincides with the long molecular axis in both cases, which is given by the intersection of the ligand planes.

**Origin of strong zero-field splitting.** It is well-established that **1** is a strongly axial system with an unusually strong ZFS[25,36]. The energy difference between the lowest two KDs is 253 cm⁻¹ in our calculations, which is very similar to the experimental excitation energy of *ca.*

235 cm⁻¹ for peak **1A** (Table 1). Projecting these states onto a pseudo-spin $\frac{3}{2}$ system[43], according to the spin Hamiltonian ('model m1')

$$\hat{H}_{m1} = D\hat{S}_z^2 + E\left(\hat{S}_x^2 - \hat{S}_y^2\right) + \mu_B \mathbf{B} \cdot \mathbf{g} \cdot \hat{\mathbf{S}}, \quad (2)$$

with $E$ as the second rank transverse ZFS parameter, and the other parameters as introduced for Eq. (1), we obtain a value of $g_z = 3.3$ for the main magnetic axis, while the two other components are close to 2.0. The ZFS tensor is aligned with the main magnetic axes and we compute $D = -127\,\text{cm}^{-1}$ and $E = 0.5\,\text{cm}^{-1}$ (Table 2). Hence, $E$ is very small, and in the following we neglect the $E$ term. The origin of the strong ZFS has already been discussed in ref. 25 in terms of a perturbative expression stemming from ligand field theory. In this context, the very small splitting to the first excited quartet state was noted, as well as that this fact makes quantitative use of the perturbation theory expression

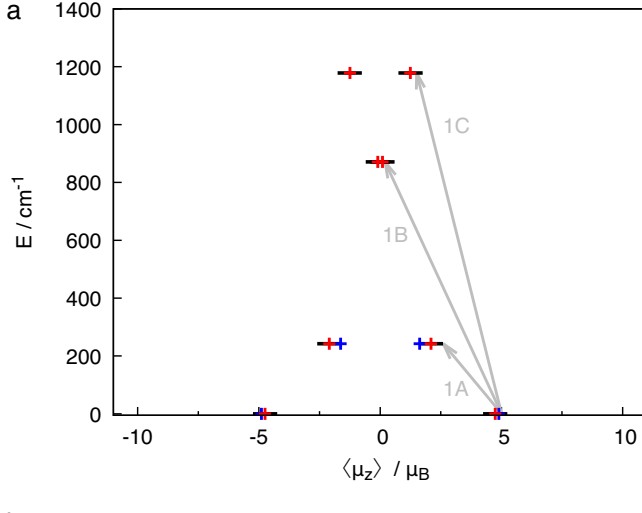

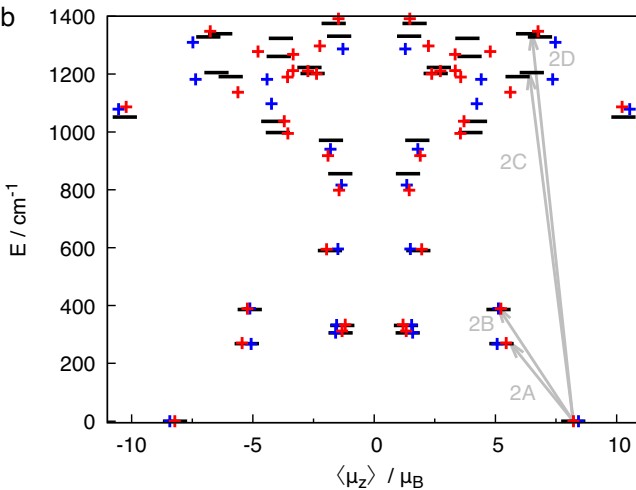

**Fig. 5 | Computed energy levels and transition energies.** Energy levels and magnetic moments along the main axis of **1** (**a**) and **2** (**b**). Ab initio energy levels are given as black horizontal bars, whilst the symbols indicate the levels obtained by fits to the spin Hamiltonians of model m1, d1 (blue) and model m2, d2 (red) discussed in the text. The gray arrows indicate the transitions that correspond to features in the experimental spectra.

questionable. The small energetic difference to the next-higher KDs is also clear from Fig. 5a, where further KDs are calculated at 847 and 1160 cm$^{-1}$. Analyzing these states at 847 cm$^{-1}$ and 1160 cm$^{-1}$ in the pseudo-spin 3/2 formalism results in parameter values of $D$ = +156 cm$^{-1}$, $E$ = 0.5 cm$^{-1}$ and a very small $g_z$ tensor component of $g_z$ = −0.8 ($g_x$ and $g_y$ values are again close to 2.0), which reflects the very small $\langle\mu_z\rangle$ expectation values of these states, as indicated in Fig. 5a.

The computations reveal that before inclusion of spin–orbit coupling, there are two low-lying quartet states separated by 477 cm$^{-1}$ (Fig. 6b). The spin–orbit coupling between these quartets is dominated by the $z$-component of the spin-orbit operator, which has an absolute value of 360 cm$^{-1}$. Therefore, we attribute the large ZFS of **1** to this strong first-order coupling of the two energetically close-lying quartet states. Indeed, in numerical experiments, in which this particular spin–orbit coupling matrix element was switched off, the ZFS reduced to around 30 cm$^{-1}$ with positive $D$ value for both quartet states.

Investigation of the main configurations of these two low-lying quartet states shows that they differ by the occupation of the $d_{x^2-y^2}$ and $d_{xy}$ orbitals, which are either doubly or singly occupied (see Fig. 6). This had been noted before[44,45]. The coupling of these two configurations via the spin–orbit operator results in two states with a strong orbital angular momentum along the main molecular axis, which

couples to the electron spins to give either an enhanced total magnetic momentum along the main axis or a reduced one. Because the orbital momentum is created by a hole in the $d_{x^2-y^2}/d_{xy}$ subshell, the spin−orbit coupled configuration with enhanced projection along the main axis is energetically more stable, in agreement with Hund's rules[46].

These observations suggest using an extended spin Hamiltonian (denoted 'model m2' in the following) of the form

$$\hat{H}_{\mathrm{m2}} = \sum_{k=0}^{1} |k\rangle \left( k\Delta + D_0\hat{S}_z^2 + \mu_B\mathbf{B}\cdot\mathbf{g}_k\cdot\hat{\mathbf{S}} \right)\langle k| + i|0\rangle\gamma\hat{S}_z\langle 1| - i|1\rangle\gamma\hat{S}_z\langle 0|$$

$$(3)$$

where $k$ runs over the two quartet states, and $i$ denotes the imaginary unit. $D_0$ is the residual splitting of the quartets due to second-order interaction with higher-lying states, $\Delta$ is the initial splitting between the spatial quartet states, and $\gamma$ is the (imaginary part of the) first-order spin-orbit coupling between the quartets. By using Eq. (3), the energy spectrum of **1** can be fit very accurately, see Fig. 5a; the corresponding values are listed in Table 2.

From a fit of the spin Hamiltonian parameters of Eq. (3) to the computed states the following parameters are obtained: $\Delta$ = 478 cm$^{-1}$ (energetic splitting of quartets without spin-orbit interaction), $\gamma$ = 352 cm$^{-1}$ (first-order spin-orbit coupling between the two quartet states) and $D_0$ = 15 cm$^{-1}$ (residual ZFS splitting from second-order couplings to higher states). This fit allows making the following assignments of the FIR and Raman spectra of **1**: The magnetic-field dependent signals around 230 cm$^{-1}$ (feature **1A**) are associated with the intramultiplet transition between the zero-field split KDs of the lower quartet (Fig. 5). The other features **1B** and **1C** are attributed to excitations from the ground KD of the quartet ground state to the two KDs of the the low-lying excited quartet state. In summary, the actual spectroscopic observation of transitions to the excited quartet state in the FIR and Raman spectra greatly support the model of a strong first-order interaction of two quartet states that leads to the extraordinarily strong ZFS of **1**.

**Interplay of spin-orbit coupling and exchange coupling in Compound 2.** The computed energy levels for compound **2** (Figs. 5b, S30) indicate that its electronic structure is substantially more complex than that of **1**. From these states and their corresponding magnetic moments, we can also compute the temperature-dependent magnetic susceptibility and find a very good agreement with the experimental data (Fig. 2a), demonstrating the accuracy of the computed energy levels. In addition to the salient feature of the electronic structure of **1**, which is the presence of a low-lying excited quartet that strongly mixes with the ground state by spin-orbit coupling, for **2** also strong exchange couplings between the radical bridge and the cobalt ions must be considered. As a first step, we have analyzed the exchange coupling by fitting the electronic states, computed without including spin-orbit coupling, to the Heisenberg Hamiltonian $\hat{H} = J\hat{\mathbf{S}}_1\cdot\hat{\mathbf{S}}_b + \hat{\mathbf{S}}_2\cdot\hat{\mathbf{S}}_b$. As shown in the Supplementary information Section S6.3 (Fig. S31), the computed electronic levels can be well understood as four intertwined spin ladders, of which the central two are degenerate, resulting from two quartet states on each cobalt ion coupled to a doublet state on the central bridge. Due to the inversion symmetry of the complex, the exchange coupling strength ($J$ = 316 cm$^{-1}$) of both cobalt ions to the radical must be the same, while direct exchange coupling between the cobalt sites can be neglected. The computations confirm antiferromagnetic exchange coupling leading to a ferrimagnetic sextet ($S = \frac{5}{2}$) ground state, and $S = \frac{3}{2}, \frac{1}{2}, \frac{1}{2}, \frac{3}{2}, \frac{5}{2}, \frac{7}{2}$ excited states for each spin ladder.

The computed energy levels can be fit to the basic spin Hamiltonian eq. (1), denoted 'model d1'. Comparison of the fits in Fig. 5b shows that this model fits the lowest levels up to 700 cm$^{-1}$ very accurately;

**Table 2 | Fit parameters for the spin Hamiltonians discussed in the text**

| Compound | Model | fit to | $D/\text{cm}^{-1}$ | $D_0/\text{cm}^{-1}$ | $\Delta/\text{cm}^{-1}$ | $\gamma/\text{cm}^{-1}$ | $J/\text{cm}^{-1}$ | $g_z$ | $g_{z,0}$ | $g_{z,1}$ |
|---|---|---|---|---|---|---|---|---|---|---|
| **1** | m1 | calc. | −127 | | | | | 3.29 | | |
| | m2 | calc. | | 15 | 478 | 352 | | | 4.86 | −0.86 |
| | m1 | exp.[a] | −130 | | | | | 3.43 | | |
| | m1 | exp.[b] | −113 | | | | | | | |
| | m2 | exp.[b] | | 10 | 500 | 322 | | | | |
| **2** | d1 | calc. | −131 | | | | 317 | 3.18 | | |
| | d2 | calc. | | −17 | 750 | 339 | 320 | | 4.40 | −2.12 |
| | d1 | exp.[c] | −145 | | | | 322 | 3.10 | | |
| | d1 | exp.[b] | −113 | | | | 390 | | | |

Model 'm1' refers to eq. (2), 'm2' to eq. (3), 'd1' to eq. (1), and 'd2' to Eq. (4).
[a] From ref. 36
[b] Fit to experimental energy levels (see text).
[c] Fit to temperature-dependent magnetic susceptibility curve, further parameters $g_x = g_y = 2.13$ (see text).

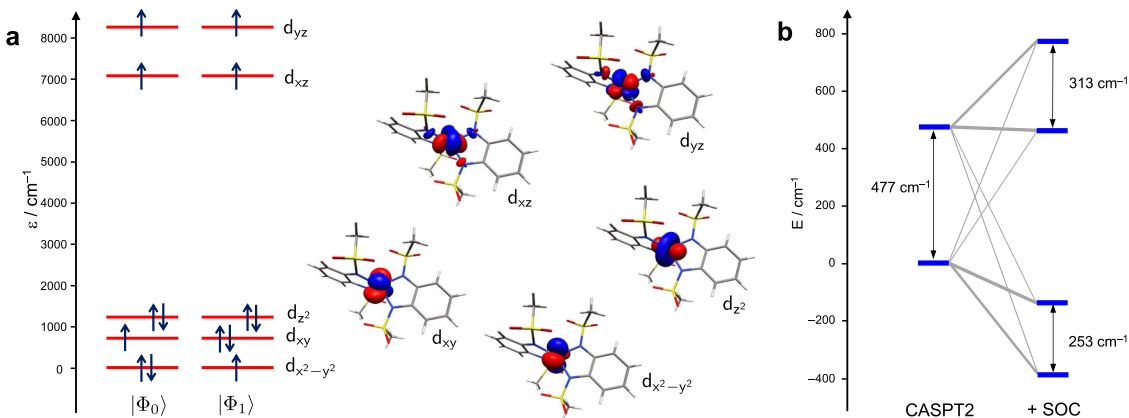

**Fig. 6 | Lowest quartet states of 1 and the effect of spin-orbit coupling.**
**a** Dominant configurations of the two lowest-lying quartet states of **1** along with the orbitals. The orbital energies are taken from the configuration-averaged Hartree-Fock computations and do not necessarily imply an aufbau-principle filling of the orbital levels. **b** Coupling of the two lowest quartet states upon including spin--orbit coupling, as computed at the MS-CASPT2 level of theory. The width of the gray lines indicates the relative contribution of the initial spatial states to the final spin--orbit coupled levels.

only for higher-lying levels around 1000 cm⁻¹ and beyond, states occur that cannot be described by model d1. Indeed, the lowest 12 states in Fig. 5b belong to the spin ladder that originates from the interaction of the lowest quartet states at each Co(II) ion with the radical bridge. The lowest six states are the components of the ground state sextet, with a strong ZFS between the $\pm\frac{5}{2}$ and $\pm\frac{3}{2}$ components. The first excited quartet state, with a relatively small positive ZFS, is found at around 400 cm⁻¹, and at around 600 cm⁻¹ a doublet follows. The ground state sextet splitting is mainly governed by the spin-orbit coupling at the cobalt centers, whereas the splitting between the different multiplets is mainly dictated by the exchange coupling (Fig. S32). Simulations show two further important points concerning the exchange coupling (Fig. S33): First, the exchange coupling strongly increases the energy barrier towards inversion of the magnetic moment, whilst at the same time suppressing tunneling rates of the magnetization. Second, they demonstrate that the value of the exchange coupling has to be large compared to the ZFS, as otherwise the effective barrier will decrease due to the presence of low-lying states generated by the exchange coupling, even for coupling strengths as strong as 100 cm⁻¹. This effect is even more prominent when three metal centers are coupled via radical ligands, hence offering an effective strategy to encounter undesired QTM.

To improve the fit especially at higher energies, the spin-orbit coupled states were fitted to a model that includes all contributions: Two quartet states on each site, separated by an energy splitting Δ, and

a spin-orbit coupling parameter γ, as previously introduced for the mononuclear species ('model m2'), as well as a strong cobalt-radical isotropic exchange coupling parameter $J$:

$$
\begin{aligned}
\hat{H}_{d2} = \sum_{k_1=0}^{1}\sum_{k_2=0}^{1} |k_1,k_2\rangle &\Big[ (k_1+k_2)\Delta + J(\hat{\mathbf{S}}_1\cdot\hat{\mathbf{S}}_b + \hat{\mathbf{S}}_2\cdot\hat{\mathbf{S}}_b) \\
&+ D_0\left(\hat{S}_{z,1}^2 + \hat{S}_{z,2}^2\right) + \mu_B\mathbf{B}\cdot\mathbf{g}_{k_1}\cdot\hat{\mathbf{S}}_1 + \mu_B\mathbf{B}\cdot\mathbf{g}_{k_2}\cdot\hat{\mathbf{S}}_2 \Big]\langle k_1,k_2| \\
&+ i|0_1\rangle\gamma\hat{S}_z\langle 1_1| + i|0_2\rangle\gamma\hat{S}_z\langle 1_2| - i|1_1\rangle\gamma\hat{S}_z\langle 0_1| - i|1_2\rangle\gamma\hat{S}_z\langle 0_2| \\
&+ \mu_B\mathbf{B}\cdot\mathbf{g}_b\cdot\hat{\mathbf{S}}_b
\end{aligned}
\tag{4}
$$

This spin Hamiltonian will be referred to as 'model d2'. Additional parameters are the residual ZFS-tensor described by $D_0$ and the g-tensors for each Co site and state, $\mathbf{g}_{k_i}$, and for the radical bridge $\mathbf{g}_b$. As in the previous 'model d1', Eq. (1), the latter is assumed to be isotropic with a fixed value of $\mathbf{g}_b = 2.0$. The $\mathbf{g}_{k_i}$ are assumed diagonal with the z-component aligned with the main magnetic axis of the overall system and the exchange coupling strength was assumed to be the same in ground and excited states of the individual ions. Only the z-components were included in the fit to the energy levels and to the $\mu_z$-values from the ab initio computations. The parameters of both cobalt centers must be the same, because the two centers are related by inversion symmetry. The fit results in Δ = 750 cm⁻¹, γ = 339 cm⁻¹ and $J$ = 320 cm⁻¹ (Table 2). The splitting of the quartet states Δ is somewhat

larger than for the mononuclear species ($\Delta = 478$ cm$^{-1}$). This is mainly an effect of the different computational procedure in the case of **2**, compared to **1**: Fewer states were included in the determination of the averaged orbitals, which was necessary in view of the large size of the system. In a computation of the analogous species in which one cobalt ion has been replaced by zinc, using the same procedure as for the mononuclear species, we find a splitting of 540 cm$^{-1}$, which is much closer to what is found in the mononuclear case. This demonstrates the limits of the accuracy of our computational procedure, but does not invalidate the overall picture. A similar deviation is found for the $D_0$ parameter, which is negative in the 'model d2' fit ($-17$ cm$^{-1}$) but positive for the mononuclear species ( $+15$ cm$^{-1}$, see above).

### Theoretical modeling of Spin-Phonon coupling

Coupling of the electron spin to local molecular vibrations is an important phenomenon that leads to relaxation of the magnetic moment (spin–lattice relaxation) in single-molecule magnets[8,30,47,48]. Spin–phonon coupling is also of importance in low-dimensional magnetism, magnetoresistance and superconductivity. Lunghi and coworkers have devoted considerable effort to spin–phonon coupling and spin–lattice relaxation in **1**, identifying which are the relevant phonons in the different temperature regimes, the importance of coherences in the density matrix, and the disentangling of Raman and Orbach processes[49–51]. Slowing down of magnetic relaxation is the main aim when improving the properties of molecular nanomagnets, intended for application in magnetic data storage. Experimentally, spectral features related to spin–phonon coupling have been observed in infrared and Raman spectra[23,25,27]. The coupling of the spin system to phonons can be modeled by a Hamiltonian of the following form (similar models were previously considered in, e.g., refs. [23,25,27,47]):

$$\hat{H} = \hat{H}_s + \hat{H}_{ph} + \hat{H}_{s-ph}. \tag{5}$$

In this expression, $\hat{H}_s$ is the pure spin Hamiltonian (e.g. Eq. (2)), while the phonons are modeled as a collection of harmonic oscillators

$$\hat{H}_{ph} = \sum_a \hbar \omega_a \left( \hat{b}_a^\dagger \hat{b}_a + \frac{1}{2} \right), \tag{6}$$

where $a$ runs over the relevant normal modes of the system and $\hat{b}_a^\dagger$ and $\hat{b}_a$ are the bosonic creation and annihilation operators associated with this mode. The coupling of the phonon modes to the spin system can be written as

$$\hat{H}_{s-ph} = \frac{1}{\sqrt{2}} \sum_{a,i,j} D_{ij}^{(a)} \hat{S}_i \hat{S}_j \left( \hat{b}_a^\dagger + \hat{b}_a \right) \tag{7}$$

where we here only consider a coupling via the ZFS tensor of the ground quartet. An expanded model would take $\hat{H}_s = \hat{H}_{m2}$, Eq. (3), and consider modulation of $D_0$ and $\gamma$ as the source of relaxation. The elements $D_{ij}^{(a)}$ are the derivatives of the ZFS tensor elements with respect to the normalized (unit-less) normal coordinate of mode $a$. Analogously to the static ZFS tensor, we define spin-phonon coupling $D^{(a)}$, and $E^{(a)}$ parameters as $D^{(a)} = \frac{3}{2} D_{zz}^{(a)}$ and $E^{(a)} = \frac{1}{2}(D_{xx}^{(a)} - D_{yy}^{(a)})$. For a quartet state, the matrix elements $\langle M, 0|\hat{H}_{s-ph}|M', 1\rangle$ read

$$
\begin{array}{c}
\begin{array}{cccc}
\quad\;\; |-\tfrac{3}{2},1\rangle & \quad |-\tfrac{1}{2},1\rangle & \quad |+\tfrac{1}{2},1\rangle & \quad |+\tfrac{3}{2},1\rangle
\end{array} \\
\begin{array}{c}
\langle -\tfrac{3}{2},0| \\
\langle -\tfrac{1}{2},0| \\
\langle +\tfrac{1}{2},0| \\
\langle +\tfrac{3}{2},0|
\end{array}
\begin{pmatrix}
D^{(a)} & \sqrt{3}(D_{xz}^{(a)} - iD_{yz}^{(a)}) & \sqrt{3}(E^{(a)} - iD_{xy}^{(a)}) & 0 \\
\sqrt{3}(D_{xz}^{(a)} + iD_{yz}^{(a)}) & -D^{(a)} & 0 & \sqrt{3}(E^{(a)} - iD_{xy}^{(a)}) \\
\sqrt{3}(E^{(a)} + iD_{xy}^{(a)}) & 0 & -D^{(a)} & -\sqrt{3}(D_{xz}^{(a)} - iD_{yz}^{(a)}) \\
0 & \sqrt{3}(E^{(a)} + iD_{xy}^{(a)}) & -\sqrt{3}(D_{xz}^{(a)} + iD_{yz}^{(a)}) & D^{(a)}
\end{pmatrix}
\end{array}
$$
$$\tag{8}$$

From this, it is clear that $D_{xz}^{(a)}$ and $D_{yz}^{(a)}$ couple states with $\Delta M = 1$ (e. g., $-\tfrac{3}{2} \leftrightarrow -\tfrac{1}{2}$), while $E^{(a)}$ and $D_{xy}^{(a)}$ couple states with $\Delta M = 2$ (e. g., $-\tfrac{3}{2} \leftrightarrow +\tfrac{1}{2}$).

In contrast, $D^{(a)}$ does not couple states with different quantum numbers $M$. To assess the influence of spin–phonon coupling on the vibrational spectra, we have calculated model spectra of a one phonon excitation coupled to a spin excitation, either close to resonance or further away at different magnetic fields (Supplementary Information, Fig. S34), considering the different spin–phonon coupling elements. In these spectra, the absorption change $\Delta A = A(B) - A(B = 0T)$ is plotted. Consequently, positive features are those where the transitions become more intense upon application of a magnetic field. These simulations reveal that indeed the $D^{(a)}$ elements do not lead to magnetic-field-dependence of the phonon spectrum, that the influence of $D_{xz}^{(a)}$ and $D_{yz}^{(a)}$ is similar to that of $E^{(a)}$ and $D_{xy}^{(a)}$ and that the effect is stronger if the uncoupled excitation energies of phonon and spin excitations are closer to resonance. For the near-resonant phonon excitation, a strongly positive signal is observed in field, i.e., the transition becomes more allowed upon application of a field. This may be understood as a consequence of a significant mixing of spin and phonon transition at zero field, which transfers some of the transition strength of the phonon to the spin-transitions, leading to a decreased phonon transition probability. As the magnetic field is switched on, the spin-transitions move to different energies, and the phonon transition gradually assumes its full, non-perturbed intensity. At frequencies further away from the zero-field spin transition in either direction, the differential signal is predominantly negative, as now the coupling to the spin system perturbs the phonon transition at finite fields, whereas it was unperturbed at zero field.

**Spin-Phonon coupling in compound 1.** An estimate of the spin-phonon coupling strength was obtained by numerical differentiation of the ZFS tensor with respect to nuclear displacements and projecting the derivative vectors onto harmonic normal modes computed at the DFT level for the isolated anion **1**. As shown in the Supplementary Information Fig. S35, the largest values occur for $D^{(a)}$ (up to 8 cm$^{-1}$) and $D_{xz}^{(a)}$ and $D_{yz}^{(a)}$ (up to 4 cm$^{-1}$), while the derivatives $E^{(a)}$ and $D_{xy}^{(a)}$ are negligible. In particular, in the region between 200 and 300 cm$^{-1}$ the values for $E^{(a)}$ and $D_{xy}^{(a)}$ are much smaller than 0.1 cm$^{-1}$, and we ignore these in the following.

The quantitative ab initio prediction of the spectra is challenging, as the relative eigenvalues of $\hat{H}_s$, and $\hat{H}_{ph}$ strongly influence the size of level splittings induced by $\hat{H}_{s-ph}$. E.g., the DFT computations of the vibrational modes likely carry errors of around 10%. The experimental FIR spectrum reveals many peaks due to vibrational excitations, and it is not feasible to assign specific vibrational modes to these peaks. Hence, to understand the main features of the FIR and Raman spectra, we have set up a simplified model. To this end, we have assumed a set of equally spaced phonon modes with $\tilde{\nu} \in \{200, 205, \ldots, 285, 290\}$ cm$^{-1}$ and equal intensities. The spin system is modeled using the simple m1 spin Hamiltonian (Eq. (2)) with $D = -113.75$ cm$^{-1}$ and $g = \text{diag}(2.0, 2.0, 3.2)$. Equal coupling strengths (due to $D_{xz}^{(a)}$) to the spin system were assumed for each of the vibrational modes. The resulting calculated FIR spectrum (Fig. 7e shares many salient features with the experimental spectrum (Fig. 7a: A strong positive feature, flanked by two weaker negative features). Note that the experimental spectrum is plotted as the zero-field transmission $Tr(B = 0$ T$)$ divided by the in-field transmission $Tr(B)$. Because absorption and transmission are related by $Tr = -\log \frac{I}{I_0}$, the sign of increased transition strength is the same for both experimental and calculated spectra. Therefore, we must conclude that the transition intensity increases upon application of a field, which is a strong indication that the experimental spectral features are indeed essentially due to electric-dipole-induced vibrational transitions that are perturbed by coupling to spin excitations. This is corroborated by the fact that the experimental features for **1A** are rather strong, which suggests that they are electric-dipole (i.e., largely vibrational), rather than magnetic-dipole (i.e., large magnetic resonance) in origin, because the transition rate of the latter is orders

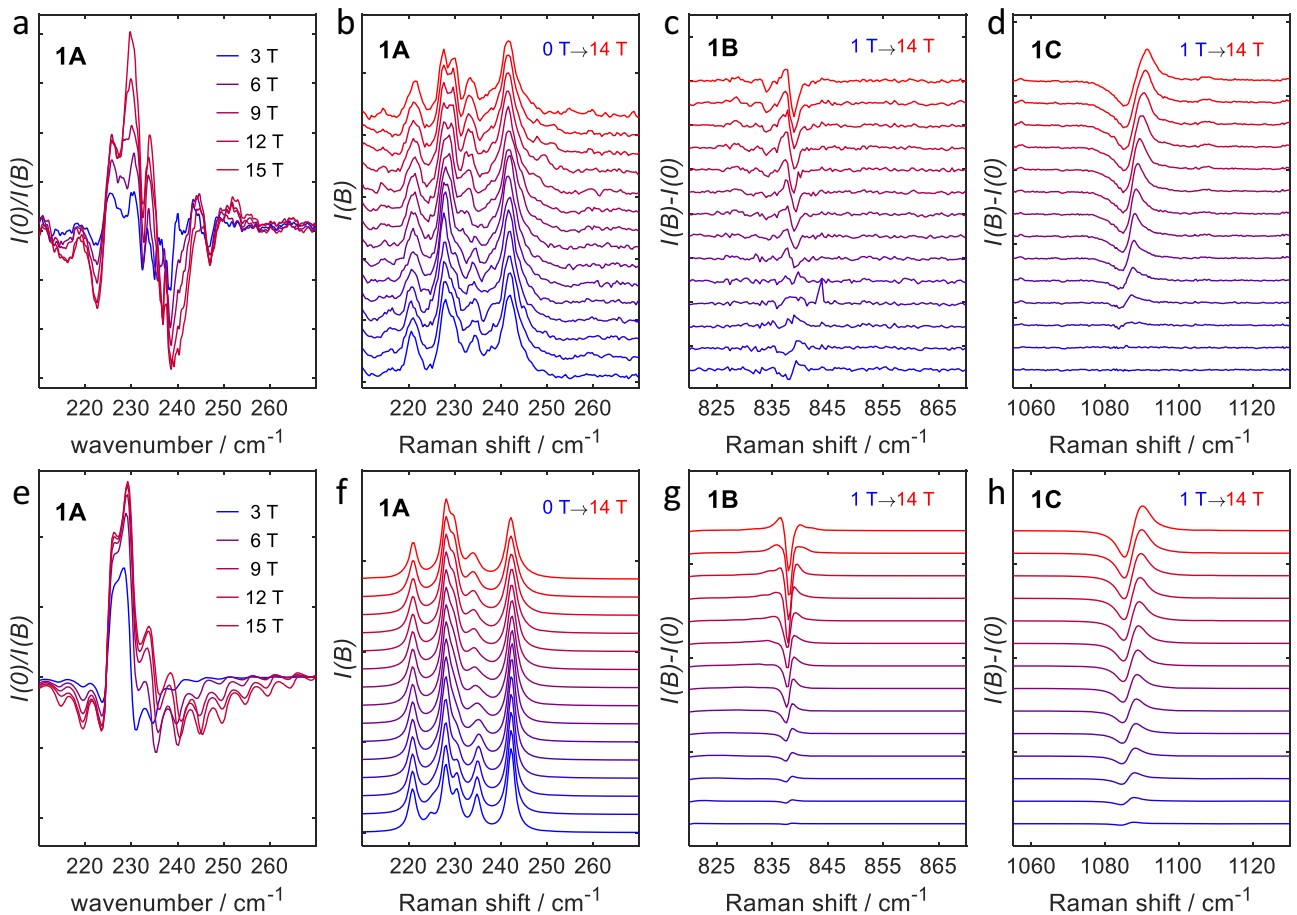

**Fig. 7 | Comparison of measured and calculated magnetic field dependent spectra of 1.** Zoom of the **1A** features, reported in Fig. 3 in the FIR and Raman spectra (**a**, **b**). The simulations of the FIR spectra are shown in **e**. The simulation of the Raman spectrum in **f** is based on five phonon modes extracted from the experiment. The magnetic field dependent Raman spectra of **1** at higher energies, shown as the difference of the in-field spectra and the 0 T spectrum, are found in **c**, **d**, **g** and **h**. The experimentally observed field dependent signals **1B** and **1C** are shown in **c** and **d** and the corresponding simulations in **g** and **h**.

of magnitude smaller than that of the former. Secondly, calculated magnetic resonance spectra (employing unphysically large transition dipole moments), using the same phonon grid model shows the opposite behavior, i.e., *decreased* absorption upon application of a magnetic field (Fig. S37). We have also tested the influence of polarization of the absorption bands, which potentially has an impact. Our simulations show that polarization only has a very small influence on the result (see Supplementary Information Section S7.4, Fig. S38) and we therefore do not consider this effect any further.

In essence, the simulations strongly suggest that the intramultiplet transition of the ground-state quartet of **1** is located in the region of the strong positive signal in Fig. 7a, i.e. between 225 and 235 cm$^{-1}$. Further details can be extracted from the same frequency region in the Raman spectrum. Because of the higher spectral quality of the Raman compared to the FIR spectra, it was possible to deconvolute the former as a sum of peaks at wavenumbers of 221, 228, 230, 234, and 242 cm$^{-1}$ with relative intensities of 0.4, 0.65, 0.6, 0.4, and 0.8. The spin-phonon coupling is again assumed to proceed via $D_{xz}^{(a)}$ with coupling strengths of 0.5, 0.0, 1.0, 1.5, and 1.0 cm$^{-1}$, and the spin system is modeled with $D = -113$ cm$^{-1}$. The resulting simulation reproduces the main features of the experimental spectrum, see Fig. 7b and f. We note in particular that the perturbation of the phonon at 234 cm$^{-1}$ has an onset at about 6 T which is in line with the strongest expected Zeeman shift (for parallel field) of the $-3/2 \rightarrow -1/2$ spin transition, see also Fig. S34a). In summary, the magnetic-field dependent FIR and Raman spectra provide us with a very accurate estimate for the axial zero-field splitting parameter of **1** of $D = -113$ cm$^{-1}$ with a conservative error margin of $\pm 2$ cm$^{-1}$.

Two further features were observed in the experimental Raman spectra (Fig. 3). Feature **1B** is located at 838 cm$^{-1}$ in zero field and was attributed to the transition from the ground KD of the ground quartet to the ground KD of the excited quartet state of the cobalt ion. In field, it shows only a very small shift of less than 1 cm$^{-1}$ to lower energies with increasing field. However, the experimental difference spectra in Fig. 7c indicate that magnetic-field-induced changes do occur, especially at field strengths above 6 T. This behavior can only be explained in terms of a phonon transition that is perturbed by spin-phonon coupling. The computations predict a very small magnetic moment of the $M = \pm\frac{1}{2}$ KD of the excited quartet state (Fig. 5), so the Zeeman shift of the spin transition is mainly due to the initial ground quartet $M = -\frac{3}{2}$ state, and, hence, the transitions to the excited quartet $M = -\frac{1}{2}$ and $M = +\frac{1}{2}$ states have very similar transition frequencies. In order to explain that a stronger magnetic-field dependent distortion only occurs for fields above approximately 6 T, we have to assume that the zero-field origin of the spin transition is located at lower energies than the phonon transition, around 820 cm$^{-1}$, and that it only couples to the phonon at 838 cm$^{-1}$. In order to include the spin–phonon coupling, we have to modify the coupling term, Eq. (7) for the 'model m2'. The coupling can be either modeled by modulation of the $xz$ or $yz$ component of the $D_0$ contribution to the ZFS ($D_{0,xz,yz}^{(a)}$), see Eq. (3), or by modulation of the spin-orbit coupling between the two quartet states (in this case, a coupling via the $x$- or $y$-component of spin-orbit coupling, $\gamma_{x,y}^{(a)}$). Both models lead to virtually the same effect. Fig. 7g shows the result of a simulation with coupling via the $x$-component of the spin–orbit coupling, assuming a value of $\gamma_x^{(a)} = 0.8$ cm$^{-1}$ for a

normalized displacement. This results (for molecules oriented along the $z$-axis) in an intersection of the spin-transitions and the phonon transition at a field strength of ca. 6 T and leads to a difference spectrum (averaged over all orientations) that is in good agreement with the experimental one.

The signal **1C** shifts more pronouncedly in field, from 1083 cm$^{-1}$ at 0 T to 1091 cm$^{-1}$ at 14 T, (Fig. 7d), and therefore differs markedly from the Raman bands at lower energy. According to the computations, excitations to the $M = \pm\frac{3}{2}$ components of the excited quartet state are expected in this region. The corresponding ground quartet $M = -\frac{3}{2} \rightarrow$ excited quartet $M = -\frac{3}{2}$ transition is spin conserving and can thus couple to the electric field of the incident radiation. We thus conclude that the signal **1C** is a purely electronic transition, without partial phonon character. The blue shift of this band indicates a smaller magnetic moment of the excited $M = -\frac{3}{2}$ state, which is in line with the predicted magnetic moments of these states in the ab initio calculations. The band can be simulated satisfactorily without assuming any additional coupling to phonons (Fig. 7h).

The observation of the three signals **1A–1C** not only corroborates our extended model of two strongly coupled quartet states, Eq. (3), it also allows extracting the essential parameters for this model from the experiment, see Table 2. In particular, the energetic splitting of the initial quartet states is $\Delta = 500$ cm$^{-1}$ and the strength of the spin-orbit coupling is $\gamma = 322$ cm$^{-1}$, in good agreement with the theoretically predicted values.

**Spin-Phonon coupling in compound 2**. As for compound **1**, the magnetic-field dependence of the FIR spectra and the low-wavenumber range of the Raman spectrum are likely due to perturbations of the phonon spectrum by spin-phonon coupling. In contrast to **1**, compound **2** possesses an inversion center, and therefore vibrations cannot be both IR- and Raman-active. As a consequence, the peaks due to spin-perturbed phonon transitions are related to different types of phonons in both types of spectra, leading to different effective transition energies. For the theoretical modeling of spin–phonon coupling in **2**, we have to extend the coupling term in the Hamiltonian to

$$\hat{H}_{\text{s-ph}} = \frac{1}{\sqrt{2}} \sum_{c=1}^{2} \sum_{a,i,j} D_{ij,c}^{(a)} \hat{S}_{i,c} \hat{S}_{j,c} \left( \hat{b}_a^\dagger + \hat{b}_a \right) \qquad (9)$$

where the index $c$ runs over the two cobalt centers. Due to the inversion symmetry of the system, the values of the derivatives at the two centers have either even or odd parity, depending on the parity of the mode indexed by $a$. Theoretical considerations suggest that the parity of the ZFS derivatives and that of the vibrational modes are the same, which is also confirmed by numerical results, see Supplementary Information Section S7.5. These numerical studies also show that the $D_{xz,c}^{(a)}$ and $D_{yz,c}^{(a)}$ derivatives are mainly responsible for spin–phonon coupling, see Supplementary Information Fig. S36. We carried out simulations of the coupling of the spin system (described by the 'model d1' spin Hamiltonian, Eq. (1)) to a grid of phonons. We focused on the transitions from the $|S\,M\rangle = |\frac{5}{2}, -\frac{5}{2}\rangle$ ground microstate to $|\frac{5}{2}, -\frac{3}{2}\rangle$ and $|\frac{5}{2}, -\frac{1}{2}\rangle$ components of the spin ground state and the $|\frac{3}{2}, -\frac{1}{2}\rangle$ and $|\frac{3}{2}, -\frac{3}{2}\rangle$ components of lowest $S = \frac{3}{2}$ excited state. We found that for even parity and coupling via $D_{xz,c}^{(a)}$, strong spin-phonon coupling is only observed for the transition to the state $|\frac{5}{2}, -\frac{3}{2}\rangle$ (**2A** in Fig. 5b), while for odd parity only the transition to $|\frac{3}{2}, -\frac{3}{2}\rangle$ (**2B** in Fig. 5b) leads to a signal. The latter has its origin in the fact that the sextet and quartet spin-orbit-coupled states have opposite parities (odd and even, respectively). Transitions to the $M = -\frac{1}{2}$ states are suppressed because $D_{xz,c}^{(a)}$ couples only to transitions with $\Delta M = \pm 1$. Because IR-allowed bands are associated with odd parity, this implies for the FIR spectra that linear spin–phonon coupling effects should only lead to non-vanishing signals for the transitions to the quartet multiplet. For

the Raman spectra, coupling to the intra-sextet transition $|\frac{5}{2}, -\frac{5}{2}\rangle \rightarrow |\frac{5}{2}, -\frac{3}{2}\rangle$ is expected. However, the FIR spectrum of **2** shows signals in both energy regions corresponding to **2A** and **2B** transitions. The peak at 240 cm$^{-1}$ is therefore assigned to a phonon transition perturbed by the $|\frac{5}{2}, -\frac{5}{2}\rangle \rightarrow |\frac{5}{2}, -\frac{3}{2}\rangle$ spin transition. We attribute the presence of a non-vanishing signal to non-linear spin-phonon coupling and/or the occurrence of anharmonicities. On the other hand, the experimental peak is clearly weaker than that observed for the mononuclear compound, potentially indicating weaker spin–phonon coupling. The signal **2B** is assigned to the phonon transition perturbed by the $|\frac{5}{2}, -\frac{5}{2}\rangle \rightarrow |\frac{3}{2}, -\frac{3}{2}\rangle$ spin transition. Because the signal is only evident for magnetic fields above 4 T, it is likely that the actual spin transition lies at somewhat lower energies (see the discussion for feature **1C** above). In Fig. 8, we have superimposed the field dependence of the spin transitions on the field-dependent FIR signals. For the modeling we assumed $D = -113$ cm$^{-1}$ (the value was adopted from the spin-Hamiltonian for **1**) and $J = 390$ cm$^{-1}$, which is bit higher than the values estimated from the SQUID measurements ($322 \pm 88$ cm$^{-1}$) and the ab initio computations (320 cm$^{-1}$). The spin Hamiltonian with these parameters gives a $|\frac{5}{2}, -\frac{5}{2}\rangle \rightarrow |\frac{5}{2}, -\frac{3}{2}\rangle$ transition at 230 cm$^{-1}$, slightly higher than $2|D|$ due to a small contribution from $J$, and a $|\frac{5}{2}, -\frac{5}{2}\rangle \rightarrow |\frac{3}{2}, -\frac{3}{2}\rangle$ transition at 393 cm$^{-1}$.

We note that the energy of the lowest transition does not significantly depend on $J$, see also Supplementary Information Fig. S32, and thus serves to provide information about the zero-field splitting in **2**. The $|\frac{5}{2}, -\frac{5}{2}\rangle \rightarrow |\frac{3}{2}, -\frac{3}{2}\rangle$ transition, on the other hand, involves components of two different multiplets of the spin-ladder and is thus directly related to the exchange coupling $J$.

In Fig. 8b, we have also overlaid the experimental Raman spectra with the computed field-dependent spin transitions from the spin-Hamiltonian model. As for the FIR spectra, no strong signals are expected for the transitions to $M = -\frac{1}{2}$ states. The field-dependence of the phonon intensities in the region 223 to 272 cm$^{-1}$ (transition **2A**) is most likely due to coupling of the phonons with the $|\frac{5}{2}, -\frac{5}{2}\rangle \rightarrow |\frac{5}{2}, -\frac{3}{2}\rangle$ transition within the sextet. As shown in the model computations in the supplementary information, the spin-transition can influence the phonon intensity, even if they are off-resonant by 20 to 30 cm$^{-1}$. The observation of a strongly field-dependent feature at 390 cm$^{-1}$, i.e., in the vicinity of the $|\frac{5}{2}, -\frac{5}{2}\rangle \rightarrow |\frac{3}{2}, -\frac{3}{2}\rangle$ transition (**2B**) contradicts the prediction that the Raman active (even parity) phonon transitions should not couple to this state in linear order (see above). Like for the FIR transitions at low energy, we have to assume that non-linear couplings and anharmonic effects in the phonon spectrum lead to this signal.

A third Raman line is observed at 917 cm$^{-1}$. This band is clearly much broader than the Raman bands observed at lower energies and the rather smooth magnetic-field-dependence suggests that this band corresponds to an electronic transition to a state of the same multiplicity as the ground state, analogously to the highest energy peak discussed for **1**. In fact, the ab initio computations show that further sextet states are expected in this energy region and so do the spin Hamiltonian models. From the simpler 'model d1', we obtain a single spin ladder which only predicts two sextets, the ground state and the excited one at $\approx 3J + |D|$. With the expected values for $J$ and $D$ this is clearly much higher than the observed excitation at 917 cm$^{-1}$. The 'model d2' (as well as the the ab initio computation) suggests additional spin ladders due to the low-lying electronic excited states at the Co centers. Transitions to these states are the more likely candidates for the assignment of the 917 cm$^{-1}$ peak. In fact, two near degenerate states are expected for the second spin ladder (see also Supplementary Information Section S6.3), but they must have different parity and the transition to only one of them is Raman active. Simulations according to the 'd2 model' imply that this assignment translates into a value of $\Delta \approx 420$ cm$^{-1}$ for the energy gap between the two single-ion quartet states, which is smaller than what is found for the mononuclear

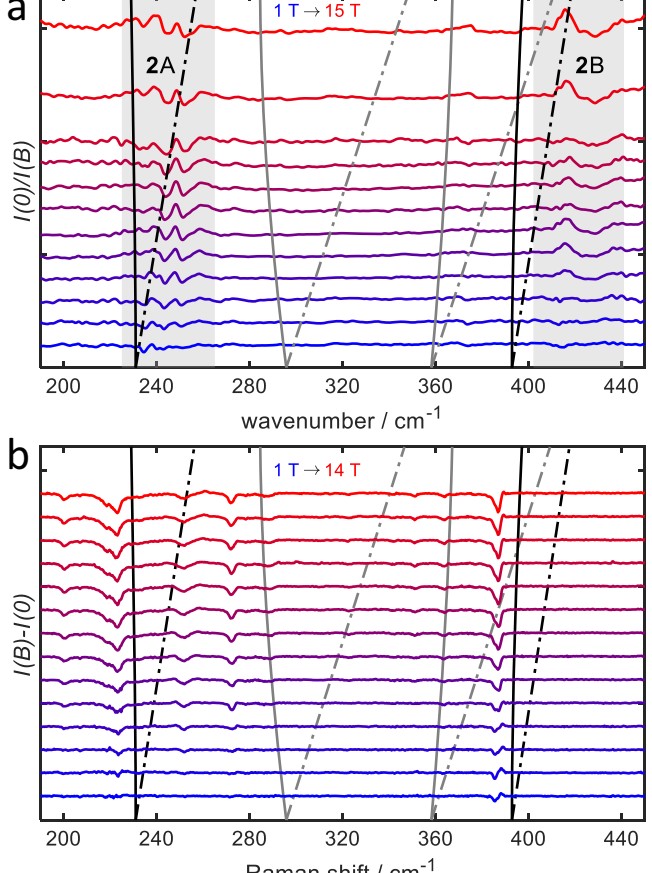

**Fig. 8 | Overlay of the calculated excitations energies of 2 with the experimentally observed ones.** Experimental FIR (**a**) and Raman (**b**) spectra of **2** at 5 K and 3 K at the indicated fields. Measurements in magnetic field were corrected by the 0 T spectrum either by division (FIR) or subtraction (Raman). The transitions energies (based on the spin Hamiltonian model 'd1' with $D = -113$ cm$^{-1}$ and $J = 390$ cm$^{-1}$) are shown as solid lines ($x$-direction) or dashed-dotted lines ($z$-direction). The transitions with $\Delta M = \pm 1$ are shown as black lines, while those with $\Delta M = \pm 2$ are shown in gray (the latter couple only weakly to phonon modes).

compound **1** ($\Delta = 500$ cm$^{-1}$). The ab initio computations do not suggest such a reduction of $\Delta$ and, therefore, predict all the sextet states at higher energies. This may be attributed to the limits of the accuracy of these computations, in particular as we have neglected spin-orbit coupling to further, higher-lying states.

The transition at 1062 cm$^{-1}$ may be attributed to a transition to the second sextet of the ground state spin ladder. From this, however, a value of only $J = 288$ cm$^{-1}$ would follow, which is in conflict with the value indicated by the position of the lowest quartet (see discussion above) and with the measured susceptibility. This likely points to the limits of the simple spin Hamiltonian models for the higher lying states, where couplings to further states may become important.

In summary, the observed FIR and Raman transitions allow us to conclude that the spin-orbit coupling at the Co centers is essentially the same as in the mononuclear compound and that the exchange coupling strength is of the order of 390 cm$^{-1}$. Additional higher-lying signals are transitions to excited sextet states and point to the presence of additional spin-ladders due to low-lying excited electron configurations at the Co centers.

In conclusion, we report a synthesis of an air-stable, radical-bridged dinuclear cobalt(II) complex, with good yields, enabling in-depth physical investigations. We have analyzed the energy spectrum of mononuclear (**1**) and radical-bridged, dinuclear (**2**) four-coordinate

cobalt(II) complexes in detail, by means of a unique combination of different spectroscopies (FIR, Raman, INS), as well as ab initio theory. Projection of ab initio energies on suitable Hamiltonians allows for direct comparison between experiment and theory. This investigation has revealed the following important points: First of all, the extraordinarily large zero-field splitting of these complexes is due to the presence of a low-lying electronic state with quartet multiplicity on the cobalt(II) ion. This excited state differs from the electronic ground state by the position of the electron hole in the $d_{xy}$ or $d_{x^2-y^2}$ orbitals. Both states mix very strongly in first order by spin−orbit coupling[45]. In contrast to previous studies, we actually observe transitions to this excited state spectroscopically, allowing the precise determination of its energy. Secondly, we have shown that the magnetic coupling in the radical bridged complex is extremely strong, and have directly observed transitions to spin-excited states. This finding shows that in this compound, spin-excited states are located at such high energies that they do not act as intermediate states in the Orbach mechanism of magnetic relaxation. In addition, the magnetization dynamics data and theoretical analyses reveal that tunneling processes are suppressed in polynuclear compared to mononuclear systems. These results underline the viability of using radical-bridged polynuclear transition metal complexes as a strategy to improve single-molecule magnet properties, such as slow relaxation of the magnetic moment. Thirdly, our investigations reveal that the magnetic-field-dependent features found in experiment are largely due to vibrational transitions that are perturbed by coupling to the spin system, rather than being due to magnetic resonance transitions. Thus, we demonstrate conclusively the importance of spin-phonon coupling for the observation of these features. Spin-phonon coupling is also at the heart of magnetic relaxation, but the majority of research in this area has so far been theoretical. Our theoretical efforts revealed that especially the off-diagonal elements $D_{xz}^{(a)}$ and $D_{yz}^{(a)}$ of the dynamic spin Hamiltonian couple the different spin microstates efficiently. This corresponds to rotation of the ZFS tensor. To move forward in terms of magnetic bistability, it appears imperative to increase the number of strongly exchange coupled ions beyond two, while keeping the single-ion ZFS tensors parallel.

## Methods
### Synthesis
**General information.** Commercially available chemicals were purchased from Sigma-Aldrich or abcr and used without further purification with the exception of [18-crown-6], which was resublimated before use.

**Synthesis of [K(18-crown-6)]$_2$[Co(bmsab)$_2$] (1).** The compound was synthesized in two steps following a previously published procedure[36]. bmsab (400 mg, 1.51 mmol, 2 equiv), Co(BF$_4$)$_2$ · 6H$_2$O (258 mg, 0.757 mmol, 1 equiv), and potassium tert-butoxide (340 mg, 3.03 mmol, 4 equiv) were added to a Schlenk flask together with MeCN (20 mL). The pink suspension was stirred for 2 days under an inert atmosphere at room temperature. The flask was then opened to air, and the solid was filtered off and washed several times with small amounts of MeCN. The pink solid was dissolved with acetone into a 500 mL round-bottomed flask, and a small amount of MeCN was added. Pink crystals of K$_2$[Co(bmsab)$_2$] were obtained by the slow diffusion of n-pentane and diethyl ether into the solution. Subsequently, KCoA (133 mg, 0.201 mmol, 1 equiv) and 18-crown-6 (106 mg, 0.402 mmol, 2 equiv) were added to a Schlenk flask together with MeCN (10 mL). The pink solution was stirred overnight under an inert atmosphere at room temperature. The solvent was removed under reduced pressure. The crude product was re-crystallized by vapor diffusion of diethyl ether into a concentrated solution of **1** in acetonitrile to give pink crystals in 80% yield (193 mg, 0.162 mmol). Purity was determined by elemental analysis.

**Anal. calcd**. for $C_{40}H_{68}CoK_2N_4O_{20}S_4$: C 40.36, H 5.76, N 4.71, S 10.77; found: C 40.37, H 5.73, N 4.63, S 10.71.

**Synthesis of Bis(bis(trimethylsilyl)amido)cobalt(II) - THF adduct.**
$Na(N(SiMe_3)_2)$ (3.67 g, 20 mmol, 2 equiv) is dissolved in THF. Anhydrous $CoCl_2$ (1.3 g, 10 mmol, 1 equiv) is suspended in THF and cooled to 0 °C. The $Na(N(SiMe_3)_2)$ solution is added dropwise via canula to the $CoCl_2$ slurry under vigorous stirring. The reaction mixture is stirred and allowed to warm to room temperature overnight, resulting in a dark green solution and copious precipitate. After the removal of volatiles, the product is extracted with 60 ml of hexane and filtered over a Celite-padded Schlenk frit. After the removal of volatiles, the crude green product is purified by sublimation ($1 \times 10^{-3}$ mbar, 75 °C) to yield the pure product as a bright green solid (2.08 g, 46%).

$^1$H NMR (250 MHz, 25 °C, $C_6D_6$) $\delta$ = 170.85 (4 H), 101.99 (4 H), −17.45 (36 H) ppm.

**Synthesis of Heteroleptic Precursor [Co(II)(bmsab)(dme)]₂.** The compound was synthesized following a published procedure[37]. A Schlenk flask is charged with finely powdered $H_2$bmsab (0.26 g, 1 mmol, 1 equiv), to which dimethoxyethane is added (10 ml). The suspension is cooled to −20 °C. $[Co(N(SiMe_3)_2)_2thf]$ (1.1 mmol, 1.1 eq.) is dissolved in dimethoxyethane (10 ml) and the solution is added dropwise to the slurry of $H_2$bmsab. The reaction mixture is brought to room temperature over the course of 1.5 h and then stirred for an additional 1.5 h, after which approximately half of the solvent is removed. Hexane (10 ml) is added and the reaction mixture stirred for 10 min. The supernatant is filtered off and the product is obtained as a fine, pink powder after drying under high vacuum for 6 h. The product is air- and moisture-sensitive and practically insoluble in its native form. Purity is determined by CHNS analysis.

**Anal. calcd**. for $C_{12}H_{20}N_2Co\,O_6S_2$: C 35.04, H 4.90, N 6.81, S 15.59; found: C 35.13, H 5.07, N 6.67, S 15.08.

**Synthesis of Bridging Ligand H₄tmsab.** The synthesis was adapted from a published procedure[52]. 1,2,4,5-Tetraminobenzene hydrochloride (2 g, 7 mmol, 1 equiv) was suspended in pyridine (50 ml). The purple suspension was cooled to 0 °C and methanesulfonyl chloride (3.2 g, 2.2 ml, 28 mmol, 4 equiv) was slowly added. The reaction mixture was warmed to room temperature and stirred for 24 h. Upon addition of 10 wt% $HCl_{aq}$ (200 ml), precipitation was observed. The brown solid was isolated by filtration, dried via suction, and subsequently washed with DCM (200 ml). The crude product was purified by suspending it in 100 ml pyridine, stirring it at 115 °C for 3 h, followed by stirring at room temperature for 72 h. After filtering off the dark red pyridine solution, a brown solid was obtained, washed with DCM and dried to yield the clean product as an off-white solid (1.65 g, 40%).$^1$H NMR (400 MHz, 25 °C, DMSO-$d_6$): $\delta$ = 3.07 (s, 12 H), 7.54 (s, 2 H), 9.04 (br s, 4 H) ppm.

**Anal. calcd**. for $C_{10}H_{18}N_4O_8S_4$: C, 26.66; H, 4.03; N, 12.44; S, 28.47. Found: C, 26.71; H, 4.073; N, 12.50; S, 28.56.

**Synthesis of [K(18-crown-6)]₂[K(H₂O)₄][Co(bmsab)₂(μ-tmsab)] (2).**
A Schlenk flask is charged with the bridging ligand $H_4$tmsab (112 mg, 0.25 mmol, 1 eq.), anhydrous KOH (56 mg, 1 mmol, 4 eq.), and 18-crown-6 (198 mg, 0.75 mmol, 3 eq.). After the addition of degassed and dry MeCN (12.5 ml), the mixture is stirred for 30 min, giving a pink suspension. The precursor [Co(bmsab)(dme)] (207 mg, 0.5 mmol, 2 eq.) is dissolved in MeCN (12.5 ml), leading to a dark red solution, and cooled to −20 °C. The suspension of the deprotonated ligand is added in five portions under the exclusion of air to give a red solution with a violet tinge. After stirring for 1.5 h, a freshly prepared solution of $FcPF_6$ (83 mg, 0.25 mmol, 1 eq.) in 5 ml MeCN is added dropwise, leading to a deep violet solution. After stirring for 1 h, the solvent is removed, giving a dark solid, which is washed with diethyl ether to remove

ferrocene. Afterwards, the product is purified by crystallization. Because the precise crystallization procedure is of great importance for the reproducible synthesis of **2**, it is described in detail in the following.

**Crystallization procedure for 2.** The crude product is dissolved in 10.5 ml deaerated MeCN with a defined water content of 20 ppt. Two NS29 Schlenk tubes (Volume of 250 ml) are then charged with seven glass test tubes (100 × 10 mm). Under argon, the crude solution is filtered through a 45 $\mu$m pore size Nylon syringe filter into the test tubes, leading to 1.5 ml per test tube. Afterward, 35 ml of diethyl ether are added to the Schlenk tube, in order to crystallize the product via vapor diffusion. After five days, large crops of dark red, rhombus-shaped platelets are obtained (140 mg, 0.075 mmol, 30%).

**Anal. calcd**. for $C_{50}H_{96}Co_2K_3N_8O_{35}S_8$: C 32.27, H 5.20, N 6.02, S 13.78; found: C 32.30, H 5.24, N 6.41, S 13.56.

**UV/Vis/NIR** ($\varepsilon$/L · cm$^{-1}$ · mol$^{-1}$): 221 (7.3 · 10$^4$), 253 (6.4 · 10$^4$), 302 (2.5 · 10$^4$), 338 (2.8 · 10$^4$), 380 (sh) (1.5 · 10$^4$), 551 (2.0 · 10$^4$), 586 (1.7 · 10$^4$), 714 (0.9 · 10$^4$) nm.

### SQUID Magnetometry

Samples were measured as pressed powders, wrapped in Teflon tape using a Quantum Design MPMS3 SQUID magnetometer. The susceptibility curves were recorded under an applied field of 1000 Oe from 1.8 to 50 K and with an applied field of 10000 Oe from 40 K to 300 K. The overlap region was then used to correct for ferromagnetic impurities. Pascal's constants were used to correct for the diamagnetic contributions[53]. The measurements on **1** were carried out on a pressed pellet of 9.6 mg sample. For the measurements on **2**, 6 mg of sample were used.

### FIR Spectroscopy

FIR spectra were either recorded on a home-built setup, using a Bruker Vertex 70v Fourier transform infrared spectrometer that is coupled to an Oxford Instruments 15T (17T) solenoid magnet or at the High Field Magnet Laboratory (HFML) in Nijmegen (Netherlands). Here, a Bruker Vertex 80v spectrometer is coupled to a 33 T Bitter magnet. In both cases, silicon bolometers were used as detectors. In the first case, an external Infrared Laboratories bolometer was used; in the second case, a home-built internal bolometer was used. The samples were prepared in both cases as 8 mm pellets. Here, a mixture of 20 weight percent of the sample inside finely ground eicosane was used.

### Raman Spectroscopy

The Raman spectrometer is a free beam setup. The 532 nm laser beam is coupled into the magnet and onto the crystal via a series of optical components and a high NA objective. A photoacoustic modulator is used to scramble the polarization of the laser beam giving unpolarized light. An ONDAX notch filter assembly was used to obtain reliable low-frequency Raman data down to 10 cm$^{-1}$. Collected scattered light was guided via an optical fiber to a spectrometer equipped with a liquid nitrogen-cooled charge-coupled device (CCD) camera. Crystalline samples of **1** and **2** were mounted on a piece of silicon wafer. The incident laser power on the sample was about 135 $\mu$W, sample temperature was around 2 K. The acquisition parameters were 5 spectra with 300 s acquisition time each.

### Inelastic neutron scattering

Inelastic neutron scattering experiments were carried out for both samples at the PANTHER spectrometer at the Institute Laue-Langevin (ILL). In both cases, non-deuterated samples were investigated. In the case of **1**, around 1 g of powdered material was placed in the neutron beam. For **2**, 500 mg were available for neutron scattering.

## Electronic structure calculations

Solid state structures were optimized using the projector augmented wave (PAW) method[54,55], as implemented in the VASP program package[56–58], with an energy cutoff for the plane-wave basis of 600 eV and the 'accurate precision' settings of VASP. The projection operators were evaluated in real space and the Brillouin zone was sampled by a $2 \times 2 \times 2$ Monkhorst-Pack grid[59]. We used the Perdew-Burke-Ernzerhof (PBE) functional[60] and a Hubbard U correction term (to enforce a high-spin configuration at each Co center) with the simplified (rotationally invariant) approach introduced by Dudarev et al.[61] with a value of 3.3 eV[62]. Dispersion corrections were applied using the zero damping DFT-D3 method[63].

The free anion structures were optimized using density functional theory as implemented in the TURBOMOLE program package[64,65]. We again employed the PBE functional, a def2-SVPD basis set[66] and the density fitting approximation[67,68], numerical integrations were done with grid '3'[69]. With the same settings also the analytic Hessian was computed[70,71], as well as the IR and the Raman intensities[72], the latter for an excitation wavelength of 532 nm and a temperature of 2 K.

The magnetic properties were calculated for the isolated anions using the the Molpro program package[73,74]. These computations employed the def2-TZVPP basis set[75] for Co and N atoms, and a def2-SVP basis set[75] for all other atoms. For the mononuclear complex we used a complete active space (CAS) with 7 electrons in 5 orbitals, denoted as CAS(7,5), and state-averaging over 40 doublet and 10 quartet states in the CAS selfconsistent field (CASSCF) calculations. For the dinuclear complex, a CAS(19,14) was employed (the d-orbitals of the two Co atoms and four $\pi$-orbitals occupied by 5 electrons of the radical bridge, see also Supplementary Information section S6.1) and state-averaging was carried out over 20 octet, 40 sextet, 40 quartet, and 40 doublet states. For the subsequent correlation and spin-orbit coupling calculations, the number of states was narrowed down to 4 octet, 8 sextet, 8 quartet, and 8 doublet states. The correlation energies of the states were obtained by a multistate pair-natural-orbital CAS second-order perturbation theory (PNO-CASPT2)[76,77] computation with a shift of 0.45 $E_h$.

Spin–orbit coupling was taken into account by the one-center approximation[78,79] in a SO-CI calculation based on the spin-orbital matrix elements computed for the CAS-CI states with updated diagonal elements and state interaction matrix elements from the multistate PNO-CASPT2 computation. The gauge origin for the orbital Zeeman contributions to the magnetic moment was set at the center of the molecule. The resulting states were analyzed in the pseudospin formalism[43], the post-processing and the evaluation of the spin-orbital models were achieved by a set of self-developed Python scripts.

Derivatives of the ZFS tensor were computed numerically by four-point central differences[31,80]. In the dinuclear case, these were calculated via diamagnetic substitution of one of the cobalt ions for a zinc ion and adding one more electron to the molecule to also obtain a diamagnetic ligand. The derivatives were then projected to the molecular normal modes in the energy ranges relevant for the experiments.

## Spin–Phonon coupling model simulations

The model spin Hamiltonians were implemented in a set of Python programs, which generate a matrix representation for a simple product basis. For phonons, only the $v = 0$ and $v = 1$ levels of the included modes were considered. The powder average was simulated by averaging over different orientations of the applied magnetic field using a Lebedev grid of order 15[81]. The orientation dependence of the IR transition strengths or Raman cross sections was neglected (see also Supplementary Information Section S7.1).

## Data availability

Crystallographic data for complex **1** has been previously reported[36] and is deposited at the Cambridge Crystallographic Data Centre (CCDC) under deposition number 1877368. Crystallographic data for complex **2** has been deposited at the CCDC under deposition number CCDC 2193374. Copies of the data can be obtained free of charge via https://www.ccdc.cam.ac.uk/structures/. The authors declare that the data supporting the findings of this study are available within the paper and its Supplementary Information files. Should any raw data files be needed in another format they are available from the corresponding author upon request.

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

## Acknowledgements
The authors acknowledge support by the state of Baden-Württemberg through bwHPC and the German Research Foundation (DFG) through grants INST 40/575-1 FUGG (JUSTUS 2 cluster), INST 41/991-1, DFG SL104/10-1, SA1840/9-1. A.K. acknowledges the support by the Stuttgart Center for Simulation Science (SimTech). D.H. and J.v.S acknowledge the support by the Landesgraduiertenförderung of the state of Baden-Württemberg. This work was also supported by HFML-RU/NWO-I, a member of the European Magnetic Field Laboratory (EMFL). We thank Robert Adam for providing some postprocessing tools used for the computation of derivatives of ZFS tensors, Dr. Wolfgang Frey for help with single-crystal diffraction measurements of **2**, and Dr. Dmitry Smirnov for access to the Raman spectrometer. This work is dedicated to the memory of Björn Fåk, who deceased on 25 July 2024.

## Author contributions
B.S., J.v.S., and A.K. conceived the research and supervised the work. SQUID magnetometry measurements and the data processing and evaluation of all spectroscopic data were carried out by D.H. Theoretical calculations were carried out by J.N. All synthetic work and the corresponding chemical characterizations were carried out by S.S based on previous work of U.A. and J.B. The Raman measurements were carried out by K.T. Far-infrared spectroscopy in high magnetic fields was carried out by D.H. under the guidance of H.E. Neutron scattering experiments were carried out by D.H. and J.v.S. under the guidance of B.F. Crystal structure determination was carried out by J.B., W.F., and I.H. Magnetometry measurements at milli Kelvin temperatures were carried out by M.S. under the supervision of W.W. The manuscript was written by D.H., J.N., S.S., B.S., A.K. and J.v.S. with input from all authors. All authors have given approval of the final version of the manuscript.

## Funding

## Competing interests
The authors declare no competing interests.
