## [Transparent Peer Review file · Nature Communications]

Electronic structure of mononuclear and radical-bridged dinuclear cobalt(II) single-molecule magnets

Corresponding Author: Professor Joris van Slageren

Version 0:

Reviewer comments:

Reviewer #1

(Remarks to the Author)

In this manuscript, Hunger et al. present a comprehensive and comparative analysis of a radical-bridged dicobalt complex and its mononuclear cobalt(II) counterpart. By employing a combination of magnetometry and various spectroscopic techniques, including FIR, Raman, and inelastic neutron scattering, the authors reveal crucial aspects such as low-lying excited states, magnetic transitions, and field-dependent behaviors. They emphasize the pivotal role of spin-phonon coupling in observing these phenomena. In my opinion, this work serves as a significant stepping stone in advancing the understanding and rational design of high-performance single-molecule magnets. I would be happy for recommending publication in Nature communication but with a major revision.

Below are some major concerns.

(1) Introduction of a strong magnetic exchange between these specific metal ions may lead to a significant enhancement in the performance of single-molecule magnets (SMMs), such as an increase in the blocking temperature or energy barrier. In their previous paper, the authors have demonstrated that a radical-bridging method effectively increased the blocking temperature (Angew 2019), but the energy barrier remained unchanged. The question is whether the use of radical-bridging methods can genuinely enhance the upper limit of SMMs based on such a low-coordinated metal ion in a specific coordination configuration. If not, one might argue why not just use lanthanide ions, which possess greater magnetic anisotropy, resulting in energy barriers as high as 2000 K, several times higher than those based on transition metal ions. I would expect the authors to address this point either through theoretical or experimental results presented to the readers.

(2) In any case, the magnetic hysteresis data for compound 2 appears to be the most crucial aspect, which regrettably was not included in the current manuscript. I would strongly suggest adding this part.

(3) It seems that compound 2 in this study, as well as a previously reported dimer, still exhibit a notable quantum tunneling of magnetization (QTM) effect, as evidenced by ac susceptibility and hysteresis measurements. In fact, the QTM issue prevails in most SMMs, whether mononuclear or radical-bridged compounds, underscoring the need for further discussion and the development of effective strategies to mitigate it. These points should be addressed in a revised manuscript.

(4) While deeper understanding of a known system is undoubtedly essential, on the other hand, the authors might need to provide more captivating statements about their current findings. Regarding synthesis, the authors highlighted the significantly higher yield compared to their previous dinuclear compound. Although this is true, it might not be a selling point in science.

Reviewer #2

(Remarks to the Author)

In the manuscript of NCOMMS-24-45931, the authors reported novel air-stable radical-bridged dinuclear cobalt(II) complex studied by a combination of magnetometry and spectroscopy (Far-Infrared, Raman and Inelastic Neutron Scattering). The use of many highly specialized and even unique techniques to determine and explain the presence of a high ZFS value undoubtedly constitutes an above-average value of the work. In my opinion, however, the most interesting fact is the use of radical objects to design SMM systems and the significant phenomena such spin-phonon coupling resulting from it, which may constitute a significant step in science and new strategy to improve single-molecule magnet properties.

It is obvious that plenty of work has been performed. I only feel competent to comment on the magnetic part. The editor should rely on other reviewers' opinions to assess this paper suitability for Nature Communication based on other aspects. In my opinion the magnetic discussion is not relatively deep and contains some inaccuracies and simplifications. Therefore,

the manuscript can be accepted for the Nature Comm. after major revision. Some problems should be corrected as listed below.

1. My first observations concern the method of magnetic measurement using the SQUID MPMS3 magnetometer.

- The measurement was taken in a Teflon type holder. Was a correction for the holder applied in the process of calculating magnetic data?
- A relatively small amount of sample was used for both constant (DC) and variable field measurements (AC), which, in my experience from working with SQUID, may cause some inaccuracies, inaccuracies, and even distortions (especially in comparison to the mass of the holder). This is particularly important in more specialized AC measurements.
- It is also not very clear to me that DC measurements are taken by setting different field values for different temperature ranges and then combining the measurements. What ferromagnetic contamination do the authors want to eliminate? This should be clarified.
- For the measurements were used respectively pressed powders (2) and pressed pellet. In my opinion, it is better to use a sample of a crushed crystal, which provides greater measurement reliability in relation to the crystal structure. Although, as the authors write in the experimental part "powder XRD was used to prove the phase purity of the batch samples of 2".

2. DC magnetic analysis.

- In the description of the magnetic data, the authors give the room temperature XMT values without reference to the pure spin values corresponding to a specific number of unpaired electrons. This makes it difficult for the reader to understand where the 5/2 state comes from.
- The spin Hamiltonian model chosen for the theoretical analysis of magnetic data also raises my doubts. As Authors wrote the XMT curve passes through a broad maximum Such a behavior usually is consistent with the involvement of the orbital angular momentum of the ground what should be included in the model. The presence of this effect seems to be confirmed by other studies.
- Moreover, the theoretical curve is not a good fit, especially in the low temperature range below 20K. It can be seen that at low temperatures additional phenomena occur (temporary increase and then decrease XMT value) which is not taken into account at all. The region below 8K is not adapted at all, either by spin Hamiltonian or ab-initio calculation.
- It is also noted that the authors did not include the usual M vs H measurements at low temperatures, which provides additional information about possible interactions.

3. AC magnetic analysis.

- It is a pity that AC measurements were not performed in a higher frequency range (around 1000Hz or 1500Hz). In the case of Co(II) ions, multichannel relaxation processes with different dynamics occur frequently. This is not a complaint, however.
 - The analysis of the dynamics of relaxation processes is very superficial. There are no designated parameters or characteristics of relaxation mechanisms. The authors write, however, that this will be the subject of a separate work. However, since the publication concerns the electronic structure and parameters determining the phenomenon of single molecule magnets and their potential tuning possibilities through barrier growth, in my opinion it should be included at least in a supplement.
 - Authors wrote: "In this context, both dimers show a drastically increased relaxation time in comparison to the monomer 1, which highlights the persistence of the relaxation behavior of radical bridged complexes of bmsab towards modifications of the ligand backbone on the one hand and on the other hand the potential of radical bridged structures in the context of molecular magnetism"
- From the curves of the determined relaxation times it can be seen that for dimers they occur according to completely different mechanisms than in the monomer. This sentence is therefore imprecise. Their detailed analysis and parameterization is needed.

In addition, the authors wrote: "Unfortunately, the two Co(II) compounds presented in this work are EPR silent"

What EPR experiment are you talking about, HF-EPR?

Reviewer #3

(Remarks to the Author)

Reviewer #4

(Remarks to the Author)

This manuscript provides an in-depth study on state-of-the-art cobalt-based single ion magnets by combining high-level synthetic chemistry with cutting-edge spectroscopic and theoretical work. The novelty of this work are the following:

- Modular synthesis of an air- and moisture-stable radical-bridged dinuclear complex, in which the strategy enables the preservation of the high-anisotropy of the mononuclear building blocks, and transform these into a low-coordinate, radical-

bridged polynuclear system.

- Unequivocal demonstration that the origin of the large zero-field splitting, i.e. the interaction at the origin of the energy barrier, is due to quasi-degeneracy of d_{xy} and $d_{x^2-y^2}$ orbitals. This insight gives a direct handle on tailoring the coordination sphere to engender even larger zero-field splittings.
- Direct spectroscopic observation of the excitations that are related to transitions between the quasi degenerate states by means of far-infrared and Raman spectroscopy. Such transitions have not been reported before.
- Strong agreement between theory and experiment that demonstrates the robustness of the description of the electronic structure.
- These results also show the viability of the metal–radical approach to developing polynuclear complexes, without generating additional low-lying spin states that short-cut the energy barrier and prevent magnetic bistability.
- Clear signatures of spin-phonon coupling in FIR and Raman spectra, which is an important driver of magnetic relaxation. The combined spectroscopic and theoretical approach presented here will have important applications for the analysis of such spectra that are increasingly being used in the community, in terms of the precise determination of spin-excitation energies.

Version 1:

Reviewer comments:

Reviewer #1

(Remarks to the Author)

I do not have any further question, it is good enough for publication.

Reviewer #2

(Remarks to the Author)

Dear Authors

After carefully reviewing the responses and the revised version of the manuscript, I recommend it for publication in Nature Comm. Additionally, I would like to emphasize that the answers were prepared reliably, and the explanations and completed measurements fully satisfy me.

Reviewer #3

(Remarks to the Author)

REVIEWER COMMENTS

Reviewer #1 (Remarks to the Author):

In this manuscript, Hunger et al. present a comprehensive and comparative analysis of a radical-bridged dicobalt complex and its mononuclear cobalt(II) counterpart. By employing a combination of magnetometry and various spectroscopic techniques, including FIR, Raman, and inelastic neutron scattering, the authors reveal crucial aspects such as low-lying excited states, magnetic transitions, and field-dependent behaviors. They emphasize the pivotal role of spin-phonon coupling in observing these phenomena. In my opinion, this work serves as a significant stepping stone in advancing the understanding and rational design of high-performance single-molecule magnets. I would be happy for recommending publication in Nature communication but with a major revision.

RESPONSE: We thank the referee for the time they invested. We were pleased to read their positive assessment regarding the significance of our work. Furthermore, we believe most of the referee's concerns deal with the analysis of the magnetization dynamics. We have carried out substantial further measurements and included a full analysis. As a result, the investigation reported in this manuscript is now more complete.

Below are some major concerns.

(1) Introduction of a strong magnetic exchange between these specific metal ions may lead to a significant enhancement in the performance of single-molecule magnets (SMMs), such as an increase in the blocking temperature or energy barrier. In their previous paper, the authors have demonstrated that a radical-bridging method effectively increased the blocking temperature (Angew 2019), but the energy barrier remained unchanged. The question is whether the use of radical-bridging methods can genuinely enhance the upper limit of SMMs based on such a low-coordinated metal ion in a specific coordination configuration. If not, one might argue why not just use lanthanide ions, which possess greater magnetic anisotropy, resulting in energy barriers as high as 2000 K, several times higher than those based on transition metal ions. I would expect the authors to address this point either through theoretical or experimental results presented to the readers.

RESPONSE: We thank the referee for this interesting point, which requires a detailed response. The bottom line is that the use of exchange coupled species can suppress tunnelling, because of the multinuclear nature of the system and because of the higher spin of the ground state of the cluster. This is the reason why Mn12ac and such systems show sizable coercivity, whilst transition metal-based single-ion magnets do not. However, if exchange interactions are weak, i.e. much weaker than the single-ion anisotropy, the exchange coupling generates additional spin states at lower energy effectively decreasing the energy barrier. In Mn12ac and similar clusters, this was never a problem, because the single-ion anisotropy was small. However, in these four-coordinate complexes, the single-ion anisotropy is enormous. Consequently, exchange interactions need to be of the same order of magnitude or higher. The metal-radical approach is the most convenient way to achieve this, as we have shown in this paper and previous work, where relaxation times increases of factors of several hundreds were found. This approach can be extended from radical-bridged dimers to trimers to tetramers and we are currently pursuing this research. This will enable to further suppress quantum tunnelling. To substantiate this statement, we have carried out a thorough theoretical analysis, which we have now included in the manuscript. Here we have calculated the magnetic transition moments for the mono- and dinuclear complexes, as well as for a hypothetical trinuclear complex, based on the spin Hamiltonian parameters that we have derived in this study. These calculations showed that i) quantum tunnelling is suppressed by 4-5 orders of magnitude each time the nuclearity is increased, ii) the effective energy barrier for the main relaxation pathway increases strongly, as expected, with nuclearity, and iii) that the energy barrier is severely decreased if the

exchange coupling is decreased from 390 to an exemplary 100 cm⁻¹. This latter point underlines the importance of the metal-radical approach, because this is the only way to achieve the necessary strong exchange interactions. In spite of much efforts, single-molecule magnets based on lanthanides have distinct disadvantages: Although energy barriers can easily exceed a 1000 K, typical lanthanide complexes do not show sizable coercivities or remanence due to a pronounced zero-field step in the hysteresis curve. We note that it is the remanence and the magnetization lifetime at zero field that are of interest, if one thinks about practical application, not the energy barrier. Exceptions to this general absence of coercivity include pentagonal-bipyramidally coordinated dysprosium complexes, endohedral metallofullerenes dysprosocenium derivatives and radical bridged lanthanide dimers. In bipyramidal and dysprosocenium type single-ion magnets, the favourable magnetization dynamics properties derive from carefully engineered crystal-field splittings, that are not necessarily robust to structural distortions. Endohedral metallofullerenes can never be synthesized in more than milligram quantities. Both dysprosocenium and radical-bridged are very to extremely sensitive to air and moisture. Both of these are clear disadvantages going towards application. In contrast, the radical bridged cobalt(II) complexes are fully air- and moisture stable. This then is the main advantage of the present type of compounds: They are structurally robust and chemically extraordinarily stable, which will allow their surface immobilization, opening avenues toward practical application. At the same time, the poor remanence and coercivities still need to be addressed further, but we have outlined clear roadmaps towards achieving this goal.

Changes to manuscript: Aspects of the above discussion have been incorporated into the introduction and results section of the manuscript. A new figure (Fig. S31) has been included in the Supplementary Information.

(2) In any case, the magnetic hysteresis data for compound 2 appears to be the most crucial aspect, which regrettably was not included in the current manuscript. I would strongly suggest adding this part.

RESPONSE: The magnetization dynamics was not the main topic of the current manuscript, which focusses on understanding the high energy barriers and strong exchange couplings observed in this class of compounds. We agree with the referee that the characterization of the magnetization dynamics should be part of any manuscript dealing with novel single-molecule magnets. Therefore, we have now added ac susceptibility, dc relaxation, hysteresis and micro-SQUID measurements for a full magnetization dynamics characterization. This has led to the inclusion of two further coauthors, Michael Schulze and Wolfgang Wernsdorfer. These measurements show very high energy barriers and long relaxation times, but, at the same time, severely waste-restricted hysteresis curves, even at mK temperatures.

Changes to manuscript: We have performed a range of additional measurements and have included this additional data on magnetization dynamics and pertinent discussion into the manuscript.

(3) It seems that compound 2 in this study, as well as a previously reported dimer, still exhibit a notable quantum tunneling of magnetization (QTM) effect, as evidenced by ac susceptibility and hysteresis measurements. In fact, the QTM issue prevails in most SMMs, whether mononuclear or radical-bridged compounds, underscoring the need for further discussion and the development of effective strategies to mitigate it. These points should be addressed in a revised manuscript.

RESPONSE: In essence, this comment continues from the first comment. We have answered more in full there. In a nutshell, we agree with the referee that QTM-like relaxation processes are still a strong and negative factor on the properties of most SMMs. In the present case, we have outlined a strategy to suppress such processes, namely the increase of nuclearity of the system, also leading to ground states with higher spins. Our new theoretical calculations support this statement.

Changes to manuscript: We have stressed this point more clearly in the text.

(4) While deeper understanding of a known system is undoubtedly essential, on the other hand, the authors might need to provide more captivating statements about their current findings. Regarding synthesis, the authors highlighted the significantly higher yield compared to their previous dinuclear compound. Although this is true, it might not be a selling point in science.

RESPONSE: We thank the referee for their assessment of our work as essential. We believe that the work will have strong impact also beyond single-molecule magnets, because, e.g. spin-phonon coupling is a factor that is also crucial for spin-based quantum technologies, and the understanding of magnetization dynamics in correlated magnetic systems in general. Poor yields can be show stoppers in the further development of functional materials, as was the case for the compounds discussed in our previous work. No thorough spectroscopic and magnetic characterization is possible for compounds that can only be made on a few mg scale. Furthermore, the dinuclear complex related here is novel, and has never been reported before. It is just related to a previously reported compound which however can only be synthesized in a few mg scale. We note that in other fields of chemistry, e.g., fine chemicals synthesis, increasing the yield can be the sole aim of the study. Therefore, in that sense we disagree with the referee.

Changes to manuscript: We have stressed the importance of understanding of spin-phonon coupling beyond single-molecule magnetism more strongly.

Reviewer #2 (Remarks to the Author):

In the manuscript of NCOMMS-24-45931, the authors reported novel air-stable radical-bridged dinuclear cobalt(II) complex studied by a combination of magnetometry and spectroscopy (Far-Infrared, Raman and Inelastic Neutron Scattering). The use of many highly specialized and even unique techniques to determine and explain the presence of a high ZFS value undoubtedly constitutes an above-average value of the work. In my opinion, however, the most interesting fact is the use of radical objects to design SMM systems and the significant phenomena such spin-phonon coupling resulting from it, which may constitute a significant step in science and new strategy to improve single-molecule magnet properties.

It is obvious that plenty of work has been performed. I only feel competent to comment on the magnetic part. The editor should rely on other reviewers' opinions to assess this paper suitability for Nature Communication based on other aspects. In my opinion the magnetic discussion is not relatively deep and contains some inaccuracies and simplifications. Therefore, the manuscript can be accepted for the Nature Comm. after major revision. Some problems should be corrected as listed below.

RESPONSE: We thank the referee for their assessment that the manuscript can be accepted in Nature Communications. We also thank them for their careful review of the magnetic characterization. As a result, we have carried out further measurements and included data and analysis into the manuscript. We now feel that the magnetic characterization is complete and we thank the referee for flagging up this issue.

1. My first observations concern the method of magnetic measurement using the SQUID MPMS3 magnetometer.

- The measurement was taken in a Teflon type holder. Was a correction for the holder applied in the process of calculating magnetic data?

RESPONSE: All data are corrected for the sample holder response (pressed, teflon-wrapped pellets, plastic straws, carefully symmetrically mounted sample). This sample holder correction is quite small compared to the magnetic moment.

- A relatively small amount of sample was used for both constant (DC) and variable field measurements (AC), which, in my experience from working with SQUID, may cause some inaccuracies, inaccuracies, and even distortions (especially in comparison to the mass of the holder). This is particularly important in more specialized AC measurements.

RESPONSE: The referee is right that care should be taken with magnetic measurements to obtain reliable data. We have checked the individual z-scans for irregularities and measured several times on different samples to ensure reproducibility. Having said this, the compound is magnetically quite concentrated and magnetic moments are rather big at the sample mass used.

- It is also not very clear to me that DC measurements are taken by setting different field values for different temperature ranges and then combining the measurements. What ferromagnetic contamination do the authors want to eliminate? This should be clarified.

RESPONSE: This in our 25+ years of experience is standard procedure in the field, to eliminate any spurious offset in the data. The term used is "ferromagnetic impurities" but this need not necessarily be the cause of the offset. Offset values are typically and also in the present case rather small.

Changes to manuscript: No changes necessary in our opinion.

- For the measurements were used respectively pressed powders (2) and pressed pellet. In my opinion, it is better to use a sample of a crushed crystal, which provides greater measurement reliability in relation to the crystal structure. Although, as the authors write in the experimental part "powder XRD was used to prove the phase purity of the batch samples of 2".

RESPONSE: We press microcrystalline powders. There is a trade-off here: On the one hand, the properties of the magnetic material may be very sensitive to small structural changes, e.g., due to loss of lattice solvent. In such cases (e.g., spin-crossover materials), use of lightly crushed crystal samples is preferred. On the other hand, use of large crystallites of highly anisotropic materials such as single-molecule magnets may lead to torquing in magnetic fields, leading to erroneous results. Because the compounds that we study are not highly sensitive to small structural changes, but are highly anisotropic, we have preferred using powders. We stress (as the referee underlines) that phase purity was checked by powder x-ray diffractometry, so the relation between the observed properties and the crystal structure is maintained.

Changes to manuscript: No changes necessary in our opinion.

2. DC magnetic analysis.

- In the description of the magnetic data, the authors give the room temperature XMT values without reference to the pure spin values corresponding to a specific number of unpaired electrons. This makes it difficult for the reader to understand where the 5/2 state comes from.

RESPONSE: Not considering the zero-field splitting, the $S = 5/2$ ground state comes from antiferromagnetic coupling between the $S = 1/2$ radical bridge and the two $S = 3/2$ cobalt(II) ions, leading to a spin ground state value of $S_{\text{tot}} = 3/2 - 1/2 + 3/2 = 5/2$.

Changes to manuscript: We have clarified this in the text.

- The spin Hamiltonian model chosen for the theoretical analysis of magnetic data also raises my doubts. As Authors wrote the XMT curve passes through a broad maximum Such a behavior usually is consistent with the involvement of the orbital angular momentum of the ground what should be included in the model. The presence of this effect seems to be confirmed by other studies.

RESPONSE: A maximum in XT can have many reasons and is not in and of itself a proof of occurrence of (first-order) orbital angular momentum. Tetrahedrally coordinated cobalt(II) ions possess a $4A_2$ ground state; in other words, no orbital angular momentum is present. However, due to the

elongation along the S4-axis of the tetrahedron, the d_{xy} and $d_{x^2-y^2}$ orbitals are almost degenerate. These two orbitals are connected by the z-component of the orbital angular momentum operator, and this is the origin of the strong anisotropy that we observed. As we show in the manuscript, this effect can be captured in first approximation by using a standard zero-field splitting Hamiltonian. It is only when delving deeper (i.e., spectroscopy) that this approximation is not good enough and the first electronic excited states on the cobalt ions must be taken into account, including spin-orbit coupling between the two states. This is formulated in Eq. 3 for the monomer and Eq. 4 for the dimer. Hence, in the manuscript (pseudo) orbital angular momentum and first-order spin-orbit coupling are already included.

Changes to manuscript: No changes necessary.

- Moreover, the theoretical curve is not a good fit, especially in the low temperature range below 20K. It can be seen that at low temperatures additional phenomena occur (temporary increase and then decrease XMT value) which is not taken into account at all. The region below 8K is not adapted at all, either by spin Hamiltonian or ab-initio calculation.

RESPONSE: The small upturn and strong downturn of the XT product are dynamic phenomena when the magnetization relaxation timescale is similar to that of the measurement. Measurements are typically performed from low to high temperatures after zero-field cooling. At the lowest temperatures, the magnetic moment of the sample is not in thermal equilibrium with the surroundings, leading to small measured magnetic moments. This is common in high-performing (i.e., slowly relaxing) single-molecule magnets.

Changes to manuscript: We have now stated the above in the discussion of the dc susceptibility measurements.

- It is also noted that the authors did not include the usual M vs H measurements at low temperatures, which provides additional information about possible interactions.

RESPONSE: We have now carried out M vs H measurements at different temperatures and included them into the supplementary information. The magnetization curves fit well with the simulations on the basis of the spin Hamiltonian parameters of the text. For the M vs H curves at low temperatures, we see signatures of slow relaxation of the magnetization.

Changes to manuscript: Data and discussion were included as described above

3. AC magnetic analysis.

- It is a pity that AC measurements were not performed in a higher frequency range (around 1000Hz or 1500Hz). In the case of Co(II) ions, multichannel relaxation processes with different dynamics occur frequently. This is not a complaint, however.

RESPONSE: We have now performed and included extensive ac susceptibility measurements recorded in different dc fields and ac frequencies of up to 1 kHz, the maximum of our state of the art Quantum Design MPMS3 magnetometer. We did not observe any signatures (double maxima in imaginary component of the ac susceptibility) of multichannel relaxation processes.

Changes to manuscript: No changes necessary.

- The analysis of the dynamics of relaxation processes is very superficial. There are no designated parameters or characteristics of relaxation mechanisms. The authors write, however, that this will be the subject of a separate work. However, since the publication concerns the electronic structure and parameters determining the phenomenon of single molecule magnets and their potential tuning possibilities through barrier growth, in my opinion it should be included at least in a supplement.

RESPONSE: We agree with the referee that a full characterization of the magnetization dynamics should be part of any publication on novel single-molecule magnets. Therefore, we have now performed extensive investigations of the magnetization dynamics.

Changes to manuscript: We have included the data and corresponding analysis into the manuscript.

• Authors wrote: “In this context, both dimers show a drastically increased relaxation time in comparison to the monomer 1, which highlights the persistence of the relaxation behavior of radical bridged complexes of bmsab towards modifications of the ligand backbone on the one hand and on the other hand the potential of radical bridged structures in the context of molecular magnetism”

From the curves of the determined relaxation times it can be seen that for dimers they occur according to completely different mechanisms than in the monomer. This sentence is therefore imprecise. Their detailed analysis and parameterization is needed.

RESPONSE: This comment follows on from the previous one, and we have now included additional data and analysis addressing this issue, as stated above. This includes an analysis of the mechanism of relaxation of the magnetic moment.

Changes to manuscript: We have included the data and corresponding analysis into the manuscript.

In addition, the authors wrote: “Unfortunately, the two Co(II) compounds presented in this work are EPR silent”

What EPR experiment are you talking about, HFEPR?

RESPONSE: Any EPR measurements. The reason is that although the system is a Kramers system, i.e. possesses a ground state with half-integer spin, the intradoublet transition is EPR-forbidden due to the high axiality of the system and the negative zero field splitting. Interdoublet transitions are allowed, and these we observe in far-infrared spectroscopy.

Changes to manuscript: We have now stated the above in manuscript.

Reviewer #3 (Remarks to the Author):

RESPONSE: We strongly support the idea of providing guidance to early career researchers for their first steps in carrying out peer review.

Reviewer #4 (Remarks to the Author):

This manuscript provides an in-depth study on state-of-the-art cobalt-based single ion magnets by combining high-level synthetic chemistry with cutting-edge spectroscopic and theoretical work. The novelty of this work are the following:

· Modular synthesis of an air- and moisture-stable radical-bridged dinuclear complex, in which the strategy enables the preservation of the high-anisotropy of the mononuclear building blocks, and transform these into a low-coordinate, radical-bridged polynuclear system.

- Unequivocal demonstration that the origin of the large zero-field splitting, i.e. the interaction at the origin of the energy barrier, is due to quasi-degeneracy of d_{xy} and $d_{x^2-y^2}$ orbitals. This insight gives a direct handle on tailoring the coordination sphere to engender even larger zero-field splittings.
- Direct spectroscopic observation of the excitations that are related to transitions between the quasi degenerate states by means of far-infrared and Raman spectroscopy. Such transitions have not been reported before.
- Strong agreement between theory and experiment that demonstrates the robustness of the description of the electronic structure.
- These results also show the viability of the metal–radical approach to developing polynuclear complexes, without generating additional low-lying spin states that short-cut the energy barrier and prevent magnetic bistability.
- Clear signatures of spin-phonon coupling in FIR and Raman spectra, which is an important driver of magnetic relaxation. The combined spectroscopic and theoretical approach presented here will have important applications for the analysis of such spectra that are increasingly being used in the community, in terms of the precise determination of spin-excitation energies.

RESPONSE: We thank the reviewer for their positive assessment and their clear summary of the novel aspects of our work.